# Learning Trees of $\ell_0$-Minimization Problems

## Abstract

The problem of computing minimally sparse solutions of under-determined linear systems $Ax = b$ is $NP$ hard in general. Subsets with extra properties, may allow efficient algorithms, most notably problems with the restricted isometry property (RIP) can be solved by convex $\ell_1$-minimization. While these classes have been very successful, they leave out many practical applications. Alternative sub-classes, can be based on the prior information that $x = Xz$ is in the (sparse) span of some suitable matrix $X$. The prior knowledge allows us to reduce assumptions on $A$ from RIP to stable rank and by means of choosing $X$ make the classes flexible. However, in order to utilize these classes in a solver, we need explicit knowledge of $X$, which, in this paper, we learn form related samples, $A$ and $b_l$, $l = 1, \ldots$. During training, we do not know $X$ yet and need other mechanisms to circumvent the hardness of the problem. We do so by organizing the samples in a hierarchical curriculum tree with a progression from easy to harder problems.

## 1 Introduction

We consider efficiently solvable subclasses of $NP$ hard problems, signed extensions of 1-in-3-SAT at the end of the paper and sparse solutions of linear systems in its main part: For matrix $A \in \mathbb{R}^{m \times n}$ and right hand side $b \in \mathbb{R}^m$, we wish to find the sparsest solution of

$$\min_{x \in \mathbb{R}^n} \|x\|_0 \quad \text{subject to} \quad Ax = b, \tag{1}$$

where $\|x\|_0$ denotes the number of non-zero entries of $x$. In full generality, this problem is $NP$-hard Natarajan (1995); Ge et al. (2011) but as many hard problems it contains tractable subclasses. Some of these are uninteresting, at least from the perspective of sparsity, e.g. problems with zero kernel $\ker(A) = 0$ and unique solution, which renders the $\ell_0$-minimization trivial. Other tractable subclasses have been extensively studied in the literature, most notably problems that satisfy the $(s, \epsilon)$-*Restricted Isometry property (RIP)*

$$(1 - \epsilon)\|x\| \leq \|Ax\| \leq (1 + \epsilon)\|x\| \quad \text{for all } s\text{-sparse } x \in \mathbb{R}^n, \tag{2}$$

with strict requirements $\epsilon < 4/\sqrt{41} \approx 0.6246$ on the RIP constants and more generally the *null space property (NSP) of order $s$*

$$\|v_S\|_1 < \|v_{\bar{S}}\|_1 \quad \text{for all } 0 \neq v \in \ker A \text{ and } |S| \leq s,$$

where $v_S$ is the restriction of $v$ to an index set $S$ and $\bar{S}$ its complement. In both cases, the sparsest solution of (1) is found by the relaxation of the sparsity $\| \cdot \|_0$ to the convex $\| \cdot \|_1$-norm

$$\min_{x \in \mathbb{R}^n} \|x\|_1 \quad \text{subject to} \quad Ax = b,$$

see Candes et al. (2006); Donoho (2006); Candès et al. (2006); Foucart & Rauhut (2013) for details.

All of these tractable subclasses are completely rigid: A problem is either contained in the class or we are out of luck. Alternatively, there are subclasses based on prior knowledge. Trivially, if we know that the solution $x = Xz$ is in the column span of a matrix $X \in \mathbb{R}^{n \times p}$, we can simplify the search space to

$$\min_{z \in \mathbb{R}^p} \|Xz\|_0 \quad \text{subject to} \quad AXz = b,$$

which, however, is no longer a standard compressed sensing problem in the variable $z$. In order to utilize compressed sensing results, we confine $X$ to sparse matrices and consider the simpler problem to find sparse $z$

$$\min_{z \in \mathbb{R}^p} \|z\|_0 \quad \text{subject to} \quad AXz = b. \tag{3}$$

so that also the product $x = Xz$ is necessarily sparse. In general this modified problem does not provide the globally sparsest solution $x$, but does so in many scenarios: E.g. if $\|Xz\|_0$ sparse solutions are unique (which is much weaker than the RIP Foucart & Rauhut (2013)), if $X$ has sufficiently sparse columns so that cancellation of non-zero entries in their span are unlikely or the compressed sensing problem admits some extra structure as in the SAT experiments at the end of the paper.

Besides the global optima question, the variant (3) has the advantage that it can be analyzed by available compressed sensing theory. Indeed, we can uniquely recover $z$ if $AX$ is RIP, which is the case for (partially) random $X$ and only mild rank conditions on $A$, see Kasiviswanathan & Rudelson (2019) and Welper (2020; 2021) in our context.

In summary, we can define tractable and adaptable subclasses by properly chosen $X$, but the algorithms require explicit knowledge of it. Since it is implausible that we just happen to know a good $X$, we learn it. While one may try to automatically uncover interesting or useful classes $X$, in this paper, we analyze the simpler option of a teacher-student setup: The teacher knows $X$ and can generate samples from the class, i.e. compressed sensing problems consisting of the measurement matrix $A$ and a right hand side $b = Ax$. The student observes only the compressed sensing problems $(A, b)$, without having the answers $x$ and reconstructs the class. On first sight, this is a cyclic problem, where the student has to solve intractable problems to uncover a class that helps her to solve otherwise intractable problems. This conundrum is solved by differentiating the problems into easy and hard ones: The former are used during training, can be solved without prior knowledge of $X$ and have sparser $z$ than the hard problems that the student can solve after training. For details, see Welper (2021) or its summary in Section 2, included to keep this paper self contained.

**New Contributions** In Welper (2021), it is difficult to find easy problems that do not require prior knowledge. This paper addresses this issue by organizing problems classes into a tree, similar to a university curriculum. Each node is a problem class, arranged so that the hard problems on the children match the easy problems on the parent. This setup allows the student to use the child prior to learn a tree node and thus inductively iterate through the tree. We prove two main theorems stated informally as follows:

1. *Theorem 3.5, Corollary 3.6:* If a tree is learnable (see Definition 3.4), the student can learn to solve all hard problems in the tree. As for compressed sensing, this is a recovery result; the student learns the knowledge of the teacher. If this constitutes $\ell_0$ minimizers is verified separately.

2. *Theorem 4.2, Corollary 4.4:* Given several assumptions, learnable trees exist, consisting of one deterministic solution and further random solutions.

These two results mimic the theory of classical compressed sensing: 1. RIP conditions ensure sparse recovery (via $\ell_1$ minimization) and 2. RIP matrices exist (e.g. i.i.d. random matrices). Further results include the following:

3. In Welper (2021) the problem class, defined by $X$ is completely random, because it is the most favorable setup for compressed sensing. As a first step towards low randomness classes, our construction allows to embed one fully deterministic problem into the root of the curriculum tree. While a single problem is not yet practical, it shows that some determinism is permissible.

4. In Section 5, we apply the learning method to a signed generalization of $NP$ complete 1-in-3-SAT problems.

In summary, unlike traditional tractable subclasses of $\ell_0$ minimization, we aim for subclasses that are adaptable and learnable by some matrix $X$. Overlaps in the classes organized into a tree together with training

samples form each class allow a student to follow a trail from easier to harder problems and avoid the $NP$-hardness of the problems alone. See Figure 1.

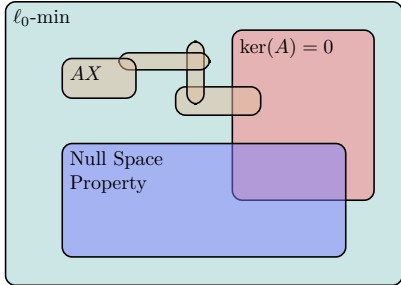

Figure 1: Tractable subclasses inside the $NP$-hard problem of sparse solutions of linear systems.

**Human Learning**   The prior knowledge informed subclasses, together with an iterative learning curriculum, are intended as a hypothetical model for human problem solving, or more concretely theorem proving.

If $P \neq NP$, and human brains have no fundamental superiority to computers, humans cannot effectively solve arbitrary instances of computationally hard problems. Yet, we routinely prove theorems and have built up a rich trove of results. But we only do so in our respective areas of expertise. Hence, one may argue that within these areas, and equipped with prior knowledge and experience, theorem proving is tractable. If so, can we program corresponding solvers into a computer? The history of artificial intelligence provides some caution. Hand coded rules in expert systems and natural language processing have proven difficult due to their immense complexity, while learned approaches are currently superior. Likewise, instead of hand crafting tractable subclasses, it seems more promising to learn them.

As a mathematical model for tractable subclasses, we consider sparse solutions of linear systems. These are $NP$-hard and in (3), we have already identified some adaptable and tractable subclasses. The solution vector $x$ is a model for a proof, as both are hard to compute. The linear combination $x = Xz$, together with the non-linear minimal sparsity, composes a candidate solution $x$ from elementary pieces in the columns of $X$, similar to assembling a proof from known tricks, techniques, lemmas and theorems.

Of course, this solution strategy is of no use if we do not know $X$. Likewise, humans need to acquire their expertise, either through training or research. An important component of both, is the solution of many related and often simplified problems. For a student, these are split into episodes, ordered by prerequisites into a curriculum tree. Likewise, for our mathematical model, we learn a tree of subclasses $X_i$ from simple samples, i.e. pairs $(A_k, b_k)$ generated form solutions in the respective classes.

As we will see (Remark 3.3), the combined knowledge of all leaf nodes $[X_1, X_2, \dots]$ in the curriculum tree is not sufficient to solve all problems in the root node $X_0$ because in an expansion $x = X_0 z_0 = \sum_i X_i z_i$, the $z_i$ combined generally have less sparsity than $z_0$ and are thus more difficult to find. Therefore, at each tree node we compress our knowledge into matrices with fewer columns and more sparse $z$. This step is similar to summarizing reoccurring proof steps into a lemma and the using it as a black box in subsequent classes.

Unlike human curricula, the model curricula in this this paper are substantially random. This is reminiscent of compressed sensing and phase retrieval, whose theory is also more random than many practical applications. In the latter areas much effort has been invested in low randomness results, yielding partially structured measurement matrices like e.g. random samples from bounded orthonormal systems. Likewise, in this paper, in an effort towards lower randomness, we allow one deterministic solution embedded into otherwise random classes. Further effort towards more realistic low randomness models is left for future work.

**Greedy Search and Heuristics**  Similar to $\ell_1$ minimization, greedy algorithms like *orthogonal matching pursuit*

$$j^{n+1} = \underset{j}{\operatorname{argmax}} \left| A_{\cdot j}^T (Ax^n - b) \right|$$
$$S^{n+1} = S^n \cup \{j^{n+1}\}$$
$$x^{n+1} = \underset{\operatorname{supp}(x) \subset S^{n+1}}{\operatorname{argmin}} \|Ax - b\|_2^2,$$

also find global $\ell_0$-minimizers under RIP assumptions Foucart & Rauhut (2013). Instead of systematically searching through an exponentially large set of candidate supports $S$, the first line provides a criterion to greedily select the next support index, based on the correlation of a column $A_{\cdot j}$ with the residual $Ax^n - b$. Applied to the modified problem (3) with prior knowledge $X$, the method changes to

$$j^{n+1} = \underset{j}{\operatorname{argmax}} \left| X_{\cdot j}^T A^T (AXz^n - b) \right|$$
$$S^{n+1} = S^n \cup \{j^{n+1}\}$$
$$z^{n+1} = \underset{\operatorname{supp}(z) \subset S^{n+1}}{\operatorname{argmin}} \|AXz - b\|_2^2.$$

In the first row, the learned knowledge $X$ modifies the index selection and thus provides a learned greedy criterion or heuristic. The learning of $X$, however, implicitly depends on a meta-heuristic as explained in Remark 3.3 below. From this perspective, the proposed methods are related to greedy and heuristic search methods in AI Russell et al. (2010); Sutton & Barto (2018); Holden (2021).

## 1.1  Related Work

**$\ell_0$-Minimization without RIP**  This paper is mainly concerned with minimally sparse solutions of systems with non-NSP or non-RIP matrices $A$. A common approach in the literature for these systems is $\ell_p$-minimization with $p < 1$, which resembles the $\ell_0$-norm more closely than the convex $\ell_1$ norm. While sparse recovery can be guaranteed for weaker variants of the RIP Candès et al. (2008); Chartrand & Staneva (2008); Foucart & Lai (2009); Sun (2012); Shen & Li (2012), these problems are again $NP$ hard Ge et al. (2011). Nonetheless, iterative solvers for $\ell_p$-minimization or non-RIP $A$ often show good results Candès et al. (2008); Chartrand & Wotao Yin (2008); Foucart & Lai (2009); Daubechies et al. (2010); Lai et al. (2013); Woodworth & Chartrand (2016).

**$\ell_0$-Minimization with Learning**  Similar to our approach, many papers study prior information for under-determined linear systems $Ax = b$. Similar to this paper, $\ell_1$ synthesis März et al. (2022) considers solutions of the form $x = Xz$, in case $x$ is not sparse in the standard basis and for random $A$. The papers Bora et al. (2017); Hand & Voroninski (2018); Huang et al. (2018); Dhar et al. (2018); Wu et al. (2019b) assume that the solution $x$ is in the range of a neural network $x = G(z; w)$, with weights pre-trained on relevant data, and then minimize $\min_z \|AG(z; w) - b\|_2$. Alternatively, the deep image prior Ulyanov et al. (2020) and compressed sensing applications Veen et al. (2020); Jagatap & Hegde (2019); Heckel & Soltanolkotabi (2020) use the architecture of an untrained network as prior and minimize the weights $\min_w \|AG(z; w) - b\|_2$ for some latent input $z$. These papers assume i.i.d. Gaussian $A$ or the Restricted Eigenvalue Condition (REC) and use the prior to select a suitable candidate among all non-unique solutions. In contrast, in the present paper, we aim for the sparsest solution and use the prior to address the hardness of the problem for difficult $A$.

The paper Wu et al. (2019a) considers an auto-encoder mechanism to find measurement matrices $A$, not only $X$, as in our case. Several other papers that combine compressed sensing with machine learning approximate the right hand side to solution map $b \to x$ by neural networks Mardani et al. (2018); Shi et al. (2017).

**Teaching Dimension**  A teacher/student setup is also considered in the *teaching dimension*. It measures how many samples a teacher needs to provide for a learner to distinguish all concepts in a concept class $\mathcal{C} \subset \{0,1\}^X$ for some finite domain $X$, see Goldman & Kearns (1995). The recursive teaching dimension

refines the idea to teaching plans, i.e. sequences of concepts and corresponding samples Zilles et al. (2011); Doliwa et al. (2014); Kirkpatrick et al. (2019). The teaching dimension is closely related to the VC-dimension Chen et al. (2016); Hu et al. (2017).

While we also learn problems in a curriculum imposing a sequential order, the goal is different: In the terminology of supervised learning, the student learns the problem to solution map $(A, b) \to x$ for $(A, b)$ is some problem class. Unlike supervised learning, this map is known to the student from the outset. The problem is rather that initially the student does not have an efficient algorithm to compute it and the learning shall help her to reduce the "problem to solution map" to a convex optimization problem.

**Knowledge Distillation**  Another area that relies on a teacher/student setup is knowledge distillation Hinton et al. (2015), where a large teacher neural network is used to train a smaller student network. See Gou et al. (2021) for an overview.

**Transfer Learning**  The progression through a tree splits the learning problem into separate episodes on different but related data sets. This is reminiscent of empirical studies on transfer- Donahue et al. (2014); Yosinski et al. (2014) and meta-learning Hospedales et al. (2020) in neural networks.

### 1.2   Notations

We use $c$ and $C$ for generic constants, independent of dimension, variance or $\psi_2$ norms that can change in each formula. We write $a \lesssim b$, $a \gtrsim b$ and $a \sim b$ for $a \leq cb$, $a \geq cb$ and $ca \leq b \leq Ca$, respectively. We denote index sets by $[n] = \{1, \ldots, n\}$ and restrictions of vectors, matrix rows and matrix columns to $J \subset [n]$ by $v_J$, $M_{J.}$ and $M_{.J}$, respectively.

## 2   Easy and Hard Problems

In this section, we summarize an easy to hard progression from Welper (2021) that allows us to progress from one node to the next, in the curriculum tree below.

### 2.1   $\ell_0$-Minimization with Prior Knowledge

For given matrix $A \in \mathbb{R}^{m \times n}$ and vector $b \in \mathbb{R}^m$, we consider the $\ell_0$-minimization problem

$$\min_{x \in \mathbb{R}^n} \|x\|_0, \quad \text{s.t.} \quad Ax = b$$

from the introduction. We have seen that this problem is $NP$-hard in general, but tractable for suitable subclasses. While the RIP and NSP conditions are rigid classes, fully determined by the matrix $A$, we now consider some more flexible ones, based on the prior knowledge that the solution is in some subset

$$\mathcal{C}_{<t} := \{x \in \mathbb{R}^n : x = Xz, \ z \text{ is } t\text{-sparse}\},$$

parametrized by some matrix $X \in \mathbb{R}^{n \times p}$ and with only mild assumptions on $A$, to be determined below.

**Remark 2.1.** *This definition does not enforce that the $x \in \mathcal{C}_{<t}$ are $\ell_0$-minimizers and likewise, the main results in this paper show recovery of $x \in \mathcal{C}_{<t}$, not $\ell_0$ minimization. In order to obtain $\ell_0$ minimizers, we need extra assumptions:*

*1. If the columns of $X$ are s-sparse, all solutions in class $\mathcal{C}_{<t}$ are st-sparse and global $\ell_0$ minimization can be guaranteed by uniqueness of st-sparse solutions. In classical compressed sensing this is implied by the RIP condition, but can also be enforced by much weaker conditions, see Foucart & Rauhut (2013).*

*2. For specific applications one may find alternative arguments. E.g. For the SAT type problems in Section 5, Lemmas 5.4 and 5.5 show global $\ell_0$ minimization of class members.*

We may regard $X$'s columns as solution components and hence assume that they are $s$-sparse, as well, for some $s > 0$, so that the solutions $x = Xz$ in class are $st$ sparse. Although the condition seems linear on first sight, the sparsity requirement of $z$ can lead to non-linear behavior as explored in detail in Welper (2021). As usual, we relax the $\ell_0$ to $\ell_1$ norm and solve the convex optimization problem

$$\min_{x \in \mathbb{R}^n} \|z\|_1, \quad \text{s.t.} \quad AXz = b. \tag{4}$$

Of course any solver requires explicit knowledge of $X$, which we discuss in detail in Section 2.2. For now, let us assume $X$ is known. Two extreme cases are noteworthy. First, without prior knowledge $X = I$, we retain standard $\ell_1$-minimization

$$\min_{x \in \mathbb{R}^n} \|x\|_1, \quad \text{s.t.} \quad Ax = b,$$

which provides correct solutions for the $\ell_0$-minimization problem if $A$ satisfies the null-space property (NSP) or the restricted isometry property (RIP), typically for sufficiently random $A$.

Second, if instead of the matrix $A$, the prior knowledge $X$ is sufficiently random, we can reduce the null-space property of $A$ to a much weaker stable rank condition on $A$. In that case, the product $AX$ satisfies a RIP with high probability (Kasiviswanathan & Rudelson (2019) and Theorem 2.5 below) and hence we can recover a unique sparse $z$. Since $X$ is also sparse, this leads to a sparse solution $x = Xz$ of the linear system $Ax = b$. In order to show that $x$ is the sparest possible solution, we need some extra structure, as discussed in Remark 2.1.

## 2.2 Learning Prior Knowledge

We have seen that subclasses $\mathcal{C}_{<t}$ of $\ell_0$-minimization problems may be tractable, given suitable prior knowledge encoded in the matrix $X$. Hence, we need a plausible model to acquire this knowledge. To this end, we consider a teacher - student scenario, with a teacher that provides sample problems and a student that infers knowledge $X$ from the samples.

The training samples must be chosen with care. Indeed, to be plausible for a variety of machine learning scenarios, we assume that the student receives samples $(A, b_i)$, but not the corresponding solutions $x_i$. On first sight, this poses a cyclic problem: We need $X$ to efficiently solve for $x_i$, but we need $x_i$ to find $X$.

To resolve this issue, we train only on a subset of easy problems $\mathcal{C}_{\text{easy}} \subset \mathcal{C}_{<t}$. These must be sufficient to fully recover $X$ and at the same time solvable by the student, without prior knowledge of $X$, by some method

$\text{SOLVE}(A, b)$:  Given an easy problem $(A, Ax)$, with $x \in \mathcal{C}_{\text{easy}}$, return $x$.

Throughout this paper, easy problems are given by $b = AXz$ for random samples $z$ with expected sparsity $\bar{t} < t$, which is strictly less than the sparsity of class $\mathcal{C}_{\leq t}$, see Assumption (A1) for details. These samples are provided by the teacher, who has access to $X$, in contrast of the student, who has not. If this class is indeed easy, depends on the existence of SOLVE and requires a delicate balance because we want the easy problems solvable but the hard ones not (otherwise training is not necessary). At this point, we do not consider the implementation of SOLVE. It will arise naturally out of the tree construction in Section 3, which also resolves the balancing issue. For comparison, the presence of easy problems may also play a role in gradient descent training of neural networks Allen-Zhu & Li (2020).

In order to recover the matrix $X$ from the easy samples $\mathcal{C}_{\text{easy}}$, the student combines the corresponding solutions into a matrix $Y$ (as columns). Since $\mathcal{C}_{\text{easy}}$ is contained in $\mathcal{C}_{<t}$, they must be of the form $Y = XZ$ for some matrix $Z$ with $t$-sparse columns. Given that $Y$ contains sufficiently many independent samples form the class $\mathcal{C}_{<t}$, sparse factorization algorithms Aharon et al. (2006); Gribonval & Schnass (2010); Spielman et al. (2012); Agarwal et al. (2014); Arora et al. (2014b;a); Neyshabur & Panigrahy (2014); Arora et al. (2015); Barak et al. (2015); Schnass (2015); Sun et al. (2017a;b); Rencker et al. (2019); Zhai et al. (2020) can recover the matrices $X$ and $Z$ up to scaling $\Gamma$ and permutation $P$.

$\text{SPARSEFACTOR}(Y)$:  Factorize $Y$ into $\bar{X} = XP\Gamma$ and $\bar{Z} = \Gamma^{-1}P^{-1}Z$ for some permutation $P$ and diagonal scaling $\Gamma$.

$\text{SCALE}$:  Scale the columns of $\bar{X}$ so that $A\bar{X}$ satisfies the *RIP*.

The permutation is irrelevant, but we need proper scaling for $\ell_1$ minimizers to work, computed by SCALE, which is a simple normalization in Welper (2021) and an application dependent function in the experiments in Section 5. We combine the discussion into TRAIN defined in Algorithm 1.

---

**Algorithm 1** Training of easy problems $\mathcal{C}_{\text{easy}}$.

---

    **function** TRAIN$(A, b_1, \ldots, b_q)$
        For all $l \in [q]$, compute $y_l = \text{SOLVE}(A, b_l)$.
        Combine all $y_l$ into the columns of a matrix $\bar{Y}$.
        Compute $\bar{X}, \bar{Z} = \text{SPARSEFACTOR}(\bar{Y})$
        **return** SCALE$(\bar{X})$.
    **end function**

---

**Remark 2.2.** *In general $\bar{Y}$ and $\bar{X}$ have the same column span and thus every $x \in \mathcal{C}_{<t}$ is given by*

$$x = \bar{X}z = \bar{Y}u.$$

*Why don't we skip the sparse factorization? While $z$ is $t$-sparse by construction, $u = Y^+ x$ is generally not. Hence, even if $Y$ is sufficiently random for $AY$ to satisfy an RIP, it is not clear that it allows us to recover $u$ by the modified $\ell_1$-minimization* (4).

### 2.3 Results

This section contains rigorous results for the algorithms of the last sections.

#### 2.3.1 Learning Prior Knowledge

We need a suitable model of random matrices, where as usual the $\psi_2$ norm is defined by $\|X\|_{\psi_2} := \sup_{p \geq 1} p^{-1/2} \mathbb{E}\left[|X|^p\right]^{1/p}$.

**Definition 2.3.** *A matrix $M \in \mathbb{R}^{n \times p}$ is $s/n$-Bernoulli-Subgaussian if $M_{jk} = \Omega_{jk} R_{jk}$, where $\Omega$ is an i.i.d. Bernoulli matrix and $R$ is an i.i.d. Subgaussian matrix with*

$$\mathbb{E}\left[\Omega_{jk}\right] = \frac{s}{n}, \quad \mathbb{E}\left[R_{jk}\right] = 0, \quad \mathbb{E}\left[R_{jk}^2\right] = \nu^2, \quad \|R_{jk}\|_{\psi_2} \leq \nu C_\psi \tag{5}$$

*for some variance $\nu > 0$. We call $M$* restricted *$s/n$ Bernoulli-Subgaussian if in addition*

$$\Pr\left[R_{jk} = 0\right] = 0, \quad \mathbb{E}\left[|R_{jk}|\right] \in \left[\frac{1}{10}, 1\right], \quad \mathbb{E}\left[R_{jk}^2\right] \leq 1, \quad \Pr\left[|R_{jk}| > \tau\right] \leq 2e^{\frac{-\tau^2}{2}}. \tag{6}$$

Next, we define the easy class $\mathcal{C}_{\text{easy}}$ as a slightly sparser version of $\mathcal{C}_{<t}$ and generate the training data by drawing random samples.

(A1) The easy class $\mathcal{C}_{\text{easy}}$ consists of solutions for $x_l = X z_l$ with columns $z_l$ of $\bar{t}/2p$ restricted Bernoulli-Subgaussian matrix $Z \in \mathbb{R}^{p \times q}$ with

$$c \log q \leq \bar{t} \leq t, \qquad\qquad q > cp^2 \log^2 p, \qquad\qquad \frac{2}{p} \leq \frac{\bar{t}}{p} \leq \frac{c}{\sqrt{p}}. \tag{7}$$

The matrix $X$ is only known to the teacher, while the student receives samples $(A, b_l)$ with $b_l = AX z_l$.

The first and last inequalities pose mild conditions on the sparsities $t$ and $\bar{t}$, while the middle inequality can always be satisfied by providing sufficiently many samples. The vectors $z_l$ have expected sparsity $\bar{t}$ and thus the corresponding solutions $X z_l$ have expected sparsity $s\bar{t}$. In order for them be easier than the full class $\mathcal{C}_{<t}$, we generally choose $\bar{t} < t$.

Next, we require the student to be accurate on easy problems, with a safety margin $\sqrt{2}$ on sparsity:

(A2) For all $\sqrt{2\bar{t}}$ sparse columns $z_l$ of $Z$, we have $\text{SOLVE}(A, AXz_l) = Xz_l$ for SOLVE as defined at the beginning of Section 2.2.

This assumption, used in Welper (2021), is delicate and will be lifted in the reminder of the paper, see Section 2.4 for more details. Since the student shall only recover the class $X$, at this point, it is not strictly necessary that the solutions $Xz_l$ are global $\ell_0$ minimizers, which can, however, be ensured by the teacher in selecting the class $X$, see Remark 2.1. Finally, we need the following technical assumption.

(A3) $X$ has full column rank.

Although this implies that $X$ has more rows than columns, that is generally not true for $AX$ used in the sparse recovery (4). The assumption results from the sparse factorization Spielman et al. (2012), where $X$ represents a basis. Newer results Agarwal et al. (2014); Arora et al. (2014b;a; 2015); Barak et al. (2015) consider over-complete bases with less rows than columns and coherence conditions and may eventually allow a weaker assumption. Anyways, as is, the assumption can be enforced by shrinking the problem class, i.e. removing columns in $X$, at the expense of being less expressive. Such a procedure is not necessary for the choices of $X$ in this paper, which have strong random components and thus full rank with high probability. With the given setup, we can recover $X$ from easy training samples as claimed in the previous sections.

**Theorem 2.4** (Welper (2021), Theorem 4.2). *Assume that (A1), (A2) and (A3) hold. Then there are constants $c > 0$ and $C \geq 0$ independent of the probability model, dimensions and sparsity, and a tractable implementation of SPARSEFACTOR (see Section 2.2) so that with probability at least*

$$1 - Cp^{-c}$$

*the output $\bar{X}$ of TRAIN (Algorithm 1) is a scaled permutation permutation $\bar{X} = XP\Gamma$ of the matrix $X$ that defines the class $\mathcal{C}_{<t}$.*

The result follows from Theorem 4.2 in Welper (2021) with some minor modifications described in Appendix A.1.

### 2.3.2 $\ell_0$-Minimization with Prior Knowledge

After we have learned $X$, we need to ensure that we can solve all problems in class $\mathcal{C}_{<t}$ by (4), not only the easy ones. We do so here for random $X$, which is clearly a idealization but common in compressed sensing, phase retrieval and related fields. Section 4 makes some progress towards more realistic classes by allowing some deterministic component. For this review, we assume

(A4) The matrix $X \in \mathbb{R}^{n \times p}$ is $(s/n\sqrt{2})$-Bernoulli-Subgaussian with

$$\frac{\|A\|_F^2}{\|A\|^2} \geq CC_\psi^4 \frac{nt}{s\epsilon^2} \log\left(\frac{3p}{\epsilon t},\right) \tag{8}$$

$\psi_2$-norm bound $C_\psi$ in the Bernoulli-Subgaussian model (5) and arbitrary constant $0 < \epsilon < 4/\sqrt{41} \approx 0.6246$ with the same bounds as the RIP constant in (2).

The left hand side $\|A\|_F^2/\|A\|^2$ is the stable rank of $A$. With the scaling

$$\text{SCALE}(\bar{X}) = \frac{\sqrt{n}}{\|A\|_F}, \tag{9}$$

we obtain the following result, with some minor modifications from the reference described in Appendix A.1.

**Theorem 2.5** (Welper (2021), Theorem 4.2). *Assume we choose (9) for SCALE and that (A1) and (A4) hold. Then there are constants $c > 0$ and $C \geq 0$ independent of the probability model, dimensions and sparsity, and a tractable implementation of SPARSEFACTOR (see Section 2.2) so that with probability at least*

$$1 - Cp^{-c}$$

*the matrix $X$ has full column rank, $s$-sparse columns and $A\bar{X}$ and satisfies the RIP*

$$(1 - \epsilon)\|v\|_2 \leq \|A\bar{X}v\|_2 \leq (1 + \epsilon)\|v\|_2 \tag{10}$$

*for all $2t$-sparse vectors $v \in \mathbb{R}^p$. Hence, for $\epsilon < 4/\sqrt{41} \approx 0.6246$, we can solve all problems in $\mathcal{C}_{<t}$ by $\ell_1$ minimization* (4).

As discussed in Remark 2.1, this is a recovery result and $\ell_0$-optimality of the class $\mathcal{C}_{<t}$ must be shown separately. In conclusion, if we train on easy samples in $\mathcal{C}_{\text{easy}}$, we can recover $X$ and thus with the modified $\ell_1$-minimization (4) solve all problems in class $\mathcal{C}_{<t}$, even the ones which we could not solve before training.

### 2.4 Implementation of the Student Solver?

While most assumptions are of technical nature the two critical ones are:

1. Implementation of SOLVE? If we implement SOLVE by plain $\ell_1$-minimization, $A$ must satisfy the $s\bar{t}$-NSP. This poses strong assumptions on $A$ and if it satisfies the slightly stronger $st$-NSP, all problems in $\mathcal{C}_{<t}$ can be solved by $\ell_1$-minimization, rendering the training of $X$ obsolete. We resolve the issue in the next section by a hierarchy of problem classes, which allow us to use prior knowledge from lower level classes to implement SOLVE.

2. Can we learn classes $X$ that are not fully random? Some partially deterministic cases are considered in Section 4.

## 3 Iterative Learning

### 3.1 Overview

We have seen that we can learn to solve all problems in a class $\mathcal{C}_{<t}$, if we are provided with samples from an easier subclass $\mathcal{C}_{\text{easy}}$. The easy class must be sufficiently rich and at the same time its sample problems must be solvable without prior training. This results in a delicate set of assumptions, which we have hidden in the existence of SOLVE, in the last section. The situation becomes much more favorable if we do not try to learn $\mathcal{C}_{<t}$ at once, but instead iteratively proceed from easy to harder and harder problems. This way, we can implement SOLVE by the outcomes of previous learning episodes, instead of uninformed plain $\ell_1$ minimizers. To this end, we order multiple problem classes into a curriculum, similar to a human student who progresses from easy to hard classes ordered by a set of prerequisites. Likewise, we consider a collection of problem classes $\mathcal{C}_i$, indexed by some index set $i \in \mathcal{I}$ and organized in a tree, e.g.

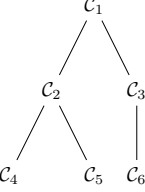

with root node $C_0$ and where each class $\mathcal{C}_i$ has finitely many children $\mathcal{C}_j$, $j \in \text{child}(i)$.

(T1) Each tree node has at most $\gamma$ children for some $\gamma \geq 0$.

The student starts learning the leafs and may proceed to a class $\mathcal{C}_i$ only if all prerequisite or child classes have been successfully learned. As before each class is given by a matrix $X_i$ with $s_i$ sparse columns and sparsity $t$

$$\mathcal{C}_i := \{x \in \mathbb{R}^n : x = X_i z, z \text{ is } t\text{-sparse}\}.$$

As in Remark 2.1, we do not enforce that class members are $\ell_0$ minimizers, which has to be ensured separately. The difficulty of each class roughly corresponds to the sparsity, with the easiest at the leafs and then less and

less sparsity towards the root of the tree. In order to learn each class $\mathcal{C}_i$, the corresponding easy problems are constructed as in Assumption (A1) in the last section

(T2) On each node $i$, the teacher provides easy problems consisting of solutions for $x_l = X_i z_l$ with columns $z_l$ of $\bar{t}/2p$ restricted Bernoulli-Subgaussian matrix $Z_i \in \mathbb{R}^{p \times q}$ with

$$c \log q \le \bar{t} \le t, \qquad q > cp^2 \log^2 p, \qquad \frac{2}{p} \le \frac{\bar{t}}{p} \le \frac{c}{\sqrt{p}}. \qquad (11)$$

The matrix $X_i$ is only known to the teacher, while the student receives samples $(A, b_l)$ with $b_l = AX_i z_l$.

Thus, the easy samples are $x = X_i z$ with random $z$ of expected sparsity $\bar{t}$. For reference, we define

$$\mathcal{C}_{\text{easy,i}} := \{x \in \mathbb{R}^n : x = X_i z, \ z \text{ is a column of } Z_i\},$$

which differs slightly from Welper (2021) and its summary in Section 2 and is only used for the following motivation.

In order to progress through the curriculum, we have to carefully connect each parent to its children. First, we assume:

(T3) The combined knowledge of all children contains the knowledge of the parent, i.e.

$$X_i = \sum_{j \in \text{child}(i)} X_j W_j =: X_{\text{child}(i)} W_{\text{child}(i)} \qquad (12)$$

for some matrices $W_j$ and combined matrices $X_{\text{child}(i)} = [X_{j_1}, \ldots, X_{j_r}]$ and $[W_{j_1}^T, \ldots, W_{j_r}^T]^T$ with $\text{child}(i) = \{j_1, \ldots, j_r\}$.

Next, we carefully calibrate the sparsity of all matrices to obtain a proper easy/hard split.

(T4) Assume that the columns of $X_i$ are $s_i$ sparse and the columns of $W_{\text{child}(i)}$ are $t/\bar{t}$ sparse with

$$t/\bar{t} \ge 1, \qquad s_i \bar{t} \le s_j t, \qquad j \in \text{child}(i). \qquad (13)$$

Then every element in the parent class satisfies $x = X_i z_i = X_{\text{child}(i)} W_{\text{child}(i)} z_i =: X_{\text{child}(i)} z_{\text{child}(i)}$. Hence, if it is easy for the parent $\|z_i\|_0 \le \bar{t}$, it is hard for the combined knowledge of the children $\|z_{\text{child}(i)}\|_0 \le t$. But given our prerequisites, we can already solve all hard children problems and implement SOLVE by the $\ell_1$ minimization (4) with prior knowledge $X_{\text{child}(i)}$. Technically, this requires that $AX_{\text{child}(i)}$ is $t$-NSP, not only all $AX_j$, $j \in \text{child}(i)$ separately. This is a relatively mild extra assumption because the NSP typically depends only logarithmically on the number of columns in $X$..

(T5) For all tree nodes $i$ the matrix product $A[\text{SCALE}(X_{\text{child}(i)})]$ as well as the root node $A[\text{SCALE}(X_0)]$ satisfy the null space property of order $\sqrt{2}t$.

With the implementation of SOLVE, we can now learn the full parent class $\mathcal{C}_i$ by Algorithm 1 and then proceed through the full tree by induction. The split (12), roughly models a set of university courses, where higher level courses recombine concepts from multiple prerequisite courses. In summary, we have the sparsities (ignoring probabilities in $\mathcal{C}_{\text{easy,i}}$)

$$
\begin{aligned}
x \in \text{Child problems} &\rightsquigarrow & x = X_{\text{child}(i)} z_{\text{child}(i)}, && \|z_{\text{child}(i)}\|_0 \le t, && \|x\|_0 \le s_j t, \\
x \in \mathcal{C}_{\text{easy,i}} &\rightsquigarrow & x = X_i z_i, && \|z_i\|_0 \le \bar{t}, && \|x\|_0 \le s_i \bar{t}, \\
x \in \mathcal{C}_i &\rightsquigarrow & x = X_i z_i, && \|z_i\|_0 \le t, && \|x\|_0 \le s_i t.
\end{aligned}
$$

It remains to learn the leafs, for which we cannot rely on any prior knowledge. To this end, note that by construction (12), we can expect the columns of the parent $X_i$ to be a factor $t/\bar{t} > 1$ less sparse than the columns of the children $X_j$, $j \in \mathrm{child}(i)$. Hence, in a carefully constructed curriculum, the tree nodes' $X_i$ become more sparse towards the bottom of the tree and ideally have unit sparsity $\mathcal{O}(1)$ at the leafs. This ensures that the leaf node classes can be solved by brute force in sub-exponential time.

(T6) We have a solver SOLVEL for the leaf nodes, satisfying Assumption (A2).

For some applications this may be costly, while for others, like SAT reductions to compressed sensing and related problems discussed in Section 5, this is routinely done for moderately sized problems Holden (2021).

We need the following two technical assumptions

(T7) For each tree node $i$, the matrix $X_i$ has full column rank.

(T8) On each tree node, we have implementations of SCALE.

These match (A3) and SCALE in Section 2, where they are discussed in more detail.

**Remark 3.1.** *All problems $x$ in class $\mathcal{C}_i$ are $t^2/\bar{t}$-sparse linear combinations of $X_{\mathrm{child}(i)}$. Hence, if $AX_{\mathrm{child}(i)}$ satisfies the $t^2/\bar{t}$ instead of only a $t$-NSP, the student can solve all problems in $\mathcal{C}_i$, without training Algorithm 1. Practically, she can jump a class, but it is increasingly difficult to jump all classes, which would render the entire learning procedure void.*

**Remark 3.2.** *The easy/hard split is achieved by some matrix satisfying a $\bar{t}$ but not a $t$ RIP. In Section 2 this matrix is $A$, so that this setup is very limiting. In this section, this is the matrix $AX_{\mathrm{child}(i)}$ and therefore at the digression of the teacher and to a large extend independent on the problem matrix $A$.*

**Remark 3.3.** *The sparse factorization in Algorithm 1 condenses the knowledge $X_{\mathrm{child}(i)}$ into $X_i$, allowing more sparse $z_i$ than $z_{\mathrm{child}(i)}$ and as a consequence to tackle more difficult, or less sparse, problems $x$. This condensation is crucial to progress in the curriculum, but is in itself a meta-heuristic to consolidate knowledge. It is comparable to Occam's razor and the human preference for simple solutions. More flexible meta-heuristics are left for future research.*

### 3.2 Learnable Trees

The algorithm of the last section is summarized in Algorithm 2 and all assumptions in the following definition.

**Definition 3.4.** *We call a tree of problem classes $\mathcal{C}_i$, $i \in \mathcal{I}$ learnable if it satisfies (T1)–(T8).*

Deferring existence of learnable trees to Section 4 below, for now we assume that a teacher has already constructed such a tree. Then, as reasoned in the last section, we can recover the knowledge $X_0$ of the root class $\mathcal{C}_0$, up to permutation and scaling in polynomial time. For a formal proof, see Appendix A.3.

**Theorem 3.5.** *Let $\mathcal{C}_i$, $i \in \mathcal{I}$ be learnable according to Definition 3.4. Then, there exists an implementation of SPARSEFACTOR and constants $c > 0$ and $C \geq 0$ independent of the probability model, dimensions and sparsity, so that with probability at least*

$$1 - C\gamma s_0^{\frac{\log \gamma}{\log(c_s t/\bar{t})}} p^{-c}$$

*the output $\bar{X}_i = \mathrm{TREETRAIN}(\mathcal{C}_i)$ of Algorithm 2 is a scaled permutation $\mathrm{SCALE}(\bar{X}_i) = \mathrm{SCALE}(X_i P)$ of $X_i$ for some permutation matrix $P$.*

Knowing all $X_i$ up to permutation and scaling allows the student to solve all problems in the tree. The proof for the following corollary is in Appendix A.3.

**Corollary 3.6.** *Assume that the event in Theorem 3.5 is true so that the student has computed* $\textsc{TreeTrain}(\mathcal{C}_i) = \bar{X}_i$ *for all three nodes* $i \in \mathcal{I}$, *which are scaled permutations* $\textsc{Scale}(\bar{X}_i) = \textsc{Scale}(X_i P)$ *of* $X_i$ *for some permutation matrix* $P$. *Then, the student can solve all problems in class* $\mathcal{C}_i$, $i \in \mathcal{I}$ *by the convex optimization problem* (3).

**Remark 3.7.** *As for compressed sensing, the last corollary is a recovery result: After training the student can find the same solutions* $x$ *in class* $\mathcal{C}_i$ *as the teacher. The corollary does not state that these are* $\ell_0$-*minimizers, which has to be verified separately. In classical compressed sensing this follows from uniqueness of sparse solutions, which is not required for the last corollary, but may be assumed in addition (and is much weaker than the RIP, see e.g. Foucart & Rauhut (2013)). Alternative verification of global* $\ell_0$ *minimization are also possible as e.g. Lemmas 5.4, 5.5 for reductions of SAT type problems to compressed sensing.*

The biggest problem with learning hard problems $\mathcal{C}_{<t}$ from easy problems $\mathcal{C}_{\mathrm{easy}}$ in Theorem 2.4 is the need for a solver for the easy problems, as discussed in Section 2.4. The hierarchical structure of Theorem 3.5 completely eradicates this assumption, except for the leaf nodes, which ideally have sparsity $\mathcal{O}(1)$ so that brute force solvers are a viable option.

---

**Algorithm 2** Tree training
$\textsc{Solve}_X$: Solve the modified $\ell_1$-minimization (4) with the given matrix $X$
$\textsc{SolveL}$: Solver for leaf nodes.
$\textsc{Train}(A, b_1, \ldots, b_q, \textsc{Solve})$: Algorithm 1 using the given solver subroutine.

   **function** $\textsc{TreeTrain}$(class $\mathcal{C}_i$)
       Get matrix $A$ and training samples $b_1, \ldots, b_q$ from teacher.
       **if** $\mathcal{C}_i$ has children **then**
           Compute $X_j = \textsc{TreeTrain}(\mathcal{C}_j)$ for $j \in \mathrm{child}(i)$
           Concatenate all child matrices $X = [X_j]_{j \in \mathrm{child}(i)}$
           **return** $X_i = \textsc{Train}(A, b_1, \ldots, b_q, \textsc{Solve}_X)$
       **else if** $\mathcal{C}_i$ has no children **then**
           **return** $X_i = \textsc{Train}(A, b_1, \ldots, b_q, \textsc{SolveL})$
       **end if**
   **end function**

---

### 3.3 Cost

Let us consider the cost of learnable trees from Definition 3.4. The number of nodes grows exponentially in the depth of the tree, but the depth only grows logarithmically with regard to the sparsity $s_0$ of the root node, given that we advance the sparsities $s_i$ as fast as (13) allows.

**Lemma 3.8.** *Let* $s_0$ *be the sparsity of the root node of the tree. Assume that each node of the tree has at most* $\gamma$ *children and that* $s_i \bar{t} \gtrsim c s_j t$ *for* $c \geq 0$ *and all* $j \in \mathrm{child}(i)$. *Then the tree has at most*

$$\gamma^{N+1} = \gamma s_0^{\frac{\log \gamma}{\log(ct/\bar{t})}}$$

*nodes.*

The proof is given in Appendix A.2. Since on each node, the number of training samples and the runtime of the training algorithm are both polynomial, this lemma ensures that the entire curriculum is learned in polynomial time, with an exponent depending on $\gamma$, and the ratio $t/\bar{t}$.

## 4 A tree Construction

In the last section, we have seen that we can learn difficult classes, given a suitable training curriculum. In this section, we argue that such curricula exist. Definition 3.4 and Theorem 3.5 state several conditions on

classes $\mathcal{C}_i$ and their matrices $X_i$ that allow the student to successfully learn the entire tree. While these are mainly simple dimensional requirements, the most severe is the NSP condition of $A[\text{SCALE}(X_{\text{child}(i)})]$. By Kasiviswanathan & Rudelson (2019) or Theorem 2.5 this is expected for random $X_i$. For a more realistic model scenario, we add a deterministic component.

The deterministic part guarantees that every global $\ell_0$-minimizer

$$\min_{x \in \mathbb{R}^n} \|x\|_0, \quad \text{s.t.} \quad Ax = b \tag{14}$$

can be embedded into a dedicated curriculum, for arbitrary right hand side $b$ and only minor rank assumptions on $A$. The random part is a placeholder for further solutions in class, to obtain a more realistic model.

**Remark 4.1.** *The model shall demonstrate that learning of any deterministic problem is possible, but is not intended as a practical curriculum design.*

### 4.1 Tree Result

Given $A$ and $x$, we construct a partially random learnable tree whose root class contains $x$ and each $X_i$ has $p$ columns for some $p > 0$. To this end, we first partition the support $\text{supp}(x)$ into non-overlapping patches $\{J_1, \ldots, J_q\} = \mathcal{J}$ and then place the corresponding sub-vectors of $x$ into $q$ columns of the matrix

$$S_{jl} := \begin{cases} x_j & j \in J_l \\ 0 & \text{else.} \end{cases} \tag{15}$$

The columns are spread into the leaf classes of the following learnable tree, were $\kappa(\cdot)$ denotes the condition number.

**Theorem 4.2.** *Let $A \in \mathbb{R}^{m \times n}$ and split $x \in \mathbb{R}^n$ into $q = 2^L$, $L \geq 1$ components $S$ given by (15). If*

1. *$AS$ has full column rank.*

2. *On each tree node, we have implementations of SCALE.*

3. *SOLVEL satisfies Assumption (A2) on the leaf nodes.*

4.

$$t \gtrsim \log p^2 + \log^3 p, \qquad\qquad 1 \lesssim t \lesssim \sqrt{p} \tag{16}$$

5.

$$\min_{J \in \mathcal{J}} \frac{\|A_{\cdot J}\|_F^2}{\|A_{\cdot J}\|^2} \gtrsim t\kappa(AS)L + t\kappa(AS) \log \frac{cqp}{t} \tag{17}$$

*for some generic constant $c$, with probability at least*

$$1 - 2\exp\left(-c\frac{1}{\kappa(AS)} \min_{J \in \mathcal{J}} \frac{\|A_{\cdot J}\|_F^2}{\|A_{\cdot J}\|^2}\right)$$

*there is a learnable binary tree of problem classes $\mathcal{C}_i$, $i \in \mathcal{I}$ of depth $L$, given by matrices $X_i$ and sparsity $t$ so that*

1. *The root class $\mathcal{C}_0$ contains $x$.*

2. *The parents are constructed from the children $X_i = X_{\text{child}(i)}W_{\text{child}(i)}$, where $W_{\text{child}(i)}$ has $t/\bar{t} = 2$ sparse columns.*

3. *The columns of the leaf nodes' $X_i$ are $|J|$ sparse.*

4. *Each class' matrix $X_i$ contains $p$ columns, consisting of columns of $S$, i.e. pieces of $x$, in the leafs and sums thereof in the interior nodes. All other entries are random (dependent between classes) or zero.*

In short, curricula that allow us to learn the root class do exist, even if we add some deterministic structure to ensure that the classes contain some meaningful result. More sophisticated classes are left for future research.

Note that $x$ can be recovered even if it is not a global $\ell_0$ minimizer. This has to be ensured separately by the designer of the curriculum, see Remarks 2.1 and 3.7. The only restriction on $x$ is Assumption 1 that $AS$ has full column rank. In case $x$ is indeed a global $\ell_0$ minimizer, this assumption is automatically satisfied by the following lemma, with $z = [1, 1, \dots]^T$. The proof is in Appendix A.4.

**Lemma 4.3.** *Assume the columns of $S \in \mathbb{R}^{n \times q}$ have non-overlapping support and $z \in \mathbb{R}^q$ with non-zero entries. If the vector $x = Sz$ is the solution of the $\ell_0$-minimization problem 14, then the columns of $AS$ are linearly independent.*

Theorem 4.2 leaves the implementation of SCALE open. The function is necessary because the sparse factorization of $Y = XZ$ into $X$ and $Z$ in Algorithm 1 is not unique up to permutation and scaling. Two options are as follows:

1. If $AX$ satisfies the RIP, all columns of $AX$ must have unit size up to the RIP constants. Hence a reasonable scaling of $X$ ensures equality $\|(AX)_{.i}\| = 1$. However, the proof only shows that $TAX$ is RIP for some preconditioner $T$, depending on the condition of the deterministic part $AS$. This implies the NSP (without preconditioning) since it is invariant under left preconditioning and hence ensures $\ell_1$ recovery. However, this is not informative for scaling $X$, unless the teacher provides the preconditioned matrix $TA$ instead of $A$.

2. The teacher can ensure that the training samples $Z$ are scaled, e.g. by sampling entries from a discrete set $\{-1, 0, 1\}$, which allows the student to recover the scaling.

Another major assumption in Theorem 4.2 is the existence of a leaf node solver SOLVEL. We can use a brute force approach if we manage to achieve enough sparsity $|J|$ in the leaf nodes, which we estimate in the following corollary.

**Corollary 4.4.** *Let $A \in \mathbb{R}^{m \times n}$. Let $x \in \mathbb{R}^n$ be a sparse, $s_x := \|x\|_0$ and for some $q$ let*

$$\mathcal{J} = \{J_1, \dots, J_q\}, \qquad\qquad \frac{s_x}{q} \le |J_i| \le 2\frac{s_x}{q}$$

*be a quasi-uniform partition of its support and $S$ be the corresponding component split defined in* (15)*.*

1. *Assume that the following sub-matrices have uniform condition number and full stable rank*

$$\kappa(AS) \lesssim 1, \qquad\qquad \frac{\|A._J\|_F^2}{\|A._J\|^2} \gtrsim |J|, \qquad\qquad J \in \mathcal{J}.$$

*with*

$$|J| \gtrsim \log s_x \log p + (\log p)^2. \tag{18}$$

2. *On each tree node, we have implementations of SCALE.*

3. *SOLVEL satisfies Assumption (A2) on the leaf nodes.*

*Then for some generic constant $c$, with probability at least*

$$1 - 2\min\left\{s_x^{-c}, p^{-c}\right\}$$

*there is a learnable binary tree of problem classes $\mathcal{C}_i$, $i \in \mathcal{I}$, given by matrices $X_i$ so that*

1. *The root class $\mathcal{C}_0$ contains $x$.*

2. *The columns of the leaf nodes' $X_i$ are $|J|$ sparse.*

3. *Each class' matrix $X_i$ contains $p$ columns*

*Proof.* We apply Theorem 4.2. Using $\kappa(AS) \lesssim 1$ and $\frac{\|A_{.J}\|_F^2}{\|A_{.J}\|^2} \gtrsim |J|$ and choosing the most favorable $t \sim \log p$ in (16), assumption (17) reduces to

$$|J| \gtrsim Lt + t\log(2^L p) \sim Lt + t\log p \sim L\log p + (\log p)^2, \tag{19}$$

posing a limit on the minimal support size we can achieve at the leafs of the tree. In order to eliminate $L$, the corollary assumes that all $J$ are of equivalent size. Since the tree has $q = 2^L$ leafs, this implies that $s_x \sim |J|2^L$ and thus $\log s_x \sim \log|J| + L \geq L$. Thus, condition (19) reduces to

$$|J| \gtrsim \log s_x \log p + (\log p)^2. \tag{20}$$

Thus, the corollary follows from Theorem 4.2 with probability at least

$$1 - 2\exp\left(-c\frac{1}{\kappa(AS)}\min_{J\in\mathcal{J}}\frac{\|A_{.J}\|_F^2}{\|A_{.J}\|^2}\right) \leq 1 - 2\exp\left(-c|J|\right),$$

for all $J \in \mathcal{J}$, which directly shows the result.

$\square$

Hence, on the leaf nodes, a brute force SolveL search of $|J|$ sparse solutions, considers about $n^{|J|} \geq n^{\log s}$ possible supports (ignoring $p$ for the time being, which is at the teachers discretion). While significantly better than $n^s$ possible supports for finding $x$ directly, the former number is not of polynomial size. In order to drive down the search size to $\mathcal{O}(1)$, we can iterate the tree construction and build new trees designed to enable the student to find every column in the leaf nodes $X_i$ with one full tree per column. At the break between curricula, this requires the teacher to provide the samples $(A, b_k)$ with $b_k = A(X_i)_{.k}$ for every leaf node column $(X_i)_{.k}$, which is a much stronger requirement than just providing arbitrary samples from the child classes in the interior nodes. Since this is more costly, we calculate in the next section that this still leads to a total tree of polynomial size.

## 4.2 Tree Extension

The curriculum in Theorem 4.2 shrinks the support size from $s$ to $\log s$. In order to reduce the size further, we may build a new curriculum for every column in every leaf $X_i$, if these columns can be split with full rank of $AS$, yielding $p2^L \leq ps$ new curricula. The assumption seems plausible for the random parts and is justified for the deterministic part by the following lemma (together with Lemma 4.3), proven in Appendix A.4.

**Lemma 4.5.** *Assume the columns of $S \in \mathbb{R}^{n \times q}$ have non-overlapping support and $z \in \mathbb{R}^q$ with non-zero entries. If the vector $x = Sz$ is the solution of the $\ell_0$-minimization problem 14, then the columns $S_{.k}$, $k \in [q]$ are global $\ell_0$ optimizers of*

$$S_{.k} \in \min_{x\in\mathbb{R}^n}\|x\|_0 \quad subject\ to \quad Ax = AS_{.k}.$$

**Remark 4.6.** *Within each curriculum, the teacher provides samples from each class. At the break between different curricula, the teacher must provide the more restrictive samples $b = Ax$ with columns $x$ of leaf node $X_i$. Weather this can be avoided in a more careful tree construction is left for future research.*

Since we aim for leaf column support size $|J| \sim 1$ and its lower bound (18) contains the number $p$ of columns in each $X_i$, which is at the teachers disposal, we shrink it together with the initial (sub-)curriculum support size $s$ by choosing $p \sim s$.

**Remark 4.7.** *By choosing a large constant in $p \sim s$ or alternatively $p \sim s^{\alpha}$, for the more difficult curricula, $p$ can be larger than $m$. But by (19), towards the simpler curricula $p$ must become small so that eventually $p \leq m$ and the matrix $AX_i$ has more rows that columns. Depending on the kernel of $AX_i$, this may void $\ell_0$ or $\ell_1$-minimization and allow simpler constructions towards the bottom of the curriculum tree.*

We iteratively repeat the procedure until the leaf support $|J| \sim \mathcal{O}(1)$ is of unit size. The total number $\#(s)$ of required (sub-)curricula for initial support size $s$ satisfies the recursive formula

$$\#(s) \sim ps\# \left( \log s \log p + (\log p)^2 \right) \geq s^2 \# \left( (\log s)^2 \right)$$

By induction, one easily verifies that $\#(s) \lesssim s^3$, so that we use only a polynomial number of curricula, each of which can be learned in polynomial time. In conclusion, combining all problem classes into one single master tree, **this yields a curriculum for a student to learn the root $\mathcal{C}_0$ in polynomial time, including a predetermined solution $x$.** The problem classes can be fairly large at the top of the tree and must be small at the leafs. At the breaks between different curricula, the training samples must be of unit size containing only one column of the next tree.

### 4.3 Construction Idea

In Theorem 4.2, all class matrices $X_i$ are derived from the single matrix

$$X := SZ^T + DR(I - ZZ^T). \tag{21}$$

The first summand is the deterministic part, with components $S$ of $x$ defined in (15) and arbitrary matrix $Z$ with sparse orthogonal columns that boosts the number of columns from $q$ to the desired $p$. The second summand is the random part with sparse random matrix $R$. The projector $(I - ZZ^T)$ ensures that it does not interfere with the deterministic part and $D$ is a scaling matrix to balance both parts.

We choose $Z$ and the support of $R$ so that, upon permutation of rows and columns $X$ is a block matrix

$$X = \begin{bmatrix} B_1 & & \\ & \ddots & \\ & & B_q \end{bmatrix}$$

with each block containing one piece $x_J$. The tree is constructed out of these blocks as follows in case $q = 4$ and analogously for larger cases.

See Appendices A.5.1 and A.6 for details.

## 5 Applications

### 5.1 3SAT and 1-in-3-SAT

For an example applications, we consider reductions from the $NP$-complete 3SAT and 1-in-3-SAT to sparse linear systems (The paper Ayanzadeh et al. (2019) considers the other direction). The problems are defined as follows.

- *Literal:* boolean variable or its negation, e.g. : $x$ or $\neg x$.

- *Clause:* disjunction of one or more literals, e.g.: $x_1 \vee \neg x_2 \vee x_3$.

- *3SAT:* satisfiability of conjunctions of clauses with three literals. For a positive result, at least one literal in each clause must be true.

- *1-in-3-SAT:* As 3SAT, but for a positive result, exactly one literal in each clause must be true.

Both problems are $NP$-complete and can easily be transformed into each other. In this section, we reduce a 1-in-3-SAT problem with clauses $c_k$, $k \in [m]$ and boolean variables $x_i$, $i \in [n]$ to a sparse linear system, following techniques from Ge et al. (2011). For each boolean variable $x_i$, we introduce two variables $y_i \in \mathbb{R}$ corresponding to $x_i$ and $z_i \in \mathbb{R}$ corresponding to $\neg x_i$ for $i \in [n]$. For each clause $c_k$, we define a pair of vectors $C_k$, $D_k$. The vector $C_k$ has a one in each entry $i$ for which the corresponding literal (not variable) $x_i$ is contained in the clause $c_k$ and likewise $D_k$ has a one in each entry $i$ for which the literal $\neg x_i$ is contained in $c_k$. All other entries of $C_k$ and $D_k$ are zero. It is easy to see that

$$y \in \{0,1\}^n \text{ and } z_i = \neg y_i$$

$$\Rightarrow \text{ Exactly one literal in } c_k \text{ is true if and only if } C_k^T y + D_k^T z = 1. \quad (22)$$

We combine the linear conditions into the linear system

$$A := \begin{bmatrix} \cdots & C_1^T & \cdots & \cdots & D_1^T & \cdots \\ & \vdots & & & \vdots & \\ \cdots & C_m^T & \cdots & \cdots & D_m^T & \cdots \\ \ddots & & & \ddots & & \\ & I_{nn} & & & I_{nn} & \\ & & \ddots & & & \ddots \end{bmatrix}, \qquad b := \begin{bmatrix} 1 \\ \vdots \\ 1 \\ 1 \\ \vdots \\ 1: \end{bmatrix} \quad (23)$$

with some extra identity blocks that together with the $\ell_0$-minimization

$$\min_{y,z \in \mathbb{R}^n} \|y\|_0 + \|z\|_0 \quad \text{subject to} \quad A \begin{bmatrix} y \\ z \end{bmatrix} = b. \quad (24)$$

ensure that $y \in \{0,1\}^n$, when possible.

**Lemma 5.1.** *The clauses $c_k$ corresponding to $C_k$ and $D_k$, $k \in [m]$ are 1-in-3 satisfiable if and only if (24) has a $n$ sparse solution.*

*Proof.* The $i$-th row of the identity blocks is $y_i + z_i = 1$. The solution is either 2-sparse or 1-sparse with $y_i = 1$, $z_i = 0$ or $y_i = 0$, $z_i = 1$. Hence the solution is at most $n$ sparse. The latter two cases are true for all $i$ if and only if $y$ and $z$ combined are $n$ sparse. Then the pair $(y_i, z_i)$ matches a boolean variable $(x_i, \neg x_i)$ and the result follows from (22).

$\square$

### 5.2 Model Class

Note that RIP proofs typically rely on random mean zero matrix entries, while reductions of random 1-in-3-SAT subclasses to compressed sensing have matrices and solution vectors with non-negative entries and thus non-zero mean. As a result, our theory is not immediately applicable. We avoid the problem by considering a larger problem class with signed solutions $x \in \mathbb{R}^n$ or $x \in \{-1, 0, 1\}^n$, which we can sample by mean-zero Bernoulli-Subgaussian distributions required for our theory. While in our 1-in-3-SAT reduction the first rows of $A$ have exactly three non-zero entries, for simplicity, we sample sparse rows

$$A = \begin{bmatrix} A_{11} & A_{12} \\ I_{n/2} & I_{n/2} \end{bmatrix} \in \mathbb{R}^{m \times n}, \qquad b = \begin{bmatrix} b_1 \\ b_2 \end{bmatrix} \in \mathbb{R}^n$$

for two sparse matrices $A_{1j} \in \{0,1\}^{(m-n/2)\times(n/2)}$. As in Lemma 5.1, the two identity blocks ensure that any solution $x$ of $Ax = b$ must have support at least $\|x\|_0 \geq \|b_2\|_0$. In the 1-in-3-SAT case, equality corresponds to satisfiable problems. Likewise, we ensure that all training problems satisfy $\|x\|_0 = \|b_2\|_0$, which automatically implies that they are global $\ell_0$ optimizers.

**Remark 5.2.** *If $\|x\|_0 = \|b_2\|_0$, then $x$ is a global $\ell_0$ minimizer.*

### 5.3 Curricula

We consider several example curricula. The first is a realization of the construction in Theorem 4.2. The following two add some extra structure to ensure global $\ell_0$ minimization properties. The Second for all columns of each $X_i$ in the curriculum and the third for all sparse linear combinations of columns of $X_i$, i.e. all elements in the corresponding problem class $\mathcal{C}_i$.

#### 5.3.1 Curriculum I

We first consider a realization of the curriculum in Theorem 4.2, as shown in Figure 2. The curriculum is constructed from a single matrix $X$ at the root node, where $*$ entries are mean-zero random $\pm 1$ and the $x$ entries are (different per entry) random $\{0,1\}$. The latter have non-zero mean, which is not amenable to RIP conditions and used as a model for the deterministic part of the theory. The children $X_i$ are constructed from the parent by zeroing out selected rows, as shown in the figure. The tree is learnable, proven in Appendix C.

**Lemma 5.3.** *Assume that the dimensions $m, n, p$, sparsities $t$, deterministic solution $x$ and matrix $A$ satisfy all assumptions of Theorem 4.2. Then the tree in Figure 2 is learnable and all other conclusions in Theorem 4.2 hold.*

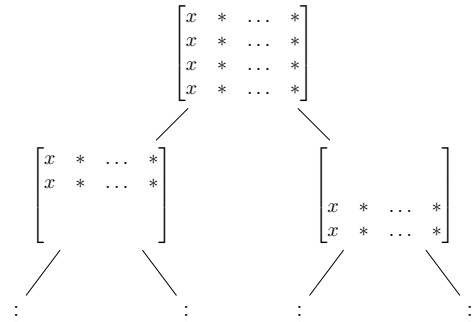

Figure 2: $X_i$ matrices for a Curriculum I. $x$ can be different in each row and $*$ are random entries.

#### 5.3.2 Curriculum II

For none of the solutions in the problem classes in Curriculum I we know if they are global $\ell_0$ minimizers. While this is not necessarily an issue for the tree construction, as outlined in Remark 3.7, it is not fully satisfactory and global minimizers can be obtained as follows. First, we split the columns according to the identity blocks in $A$, as shown in Figure 3. Each component in the upper block $y$ or $*$, has exactly one corresponding component in the lower block $z$ or $+$ so that for each pair at most one entry is non-zero. As a result each column has the required sparsity to guarantee that it is a global $\ell_0$ minimum by Remark 5.2.

**Lemma 5.4.** *For all nodes $i \in \mathcal{I}$ in Curriculum II, all columns of $X_i$ are global $\ell_0$ minimizers.*

Since the random parts of the curriculum have dependencies, we can no longer apply Theorem 4.2 to show that this tree is learnable, although it is conceivable that similar results still apply.

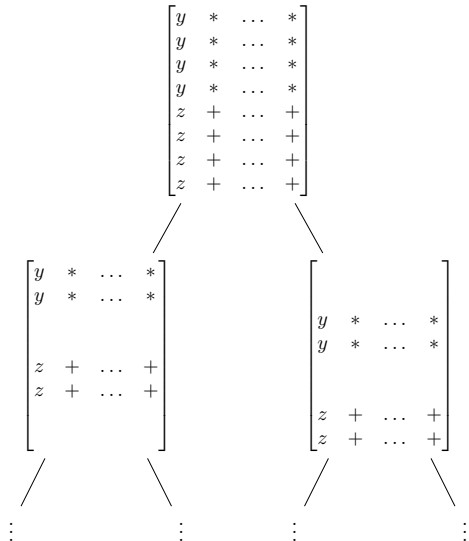

Figure 3: $X_i$ matrices for Curriculum II with $\ell_0$ minimal columns.

### 5.3.3 Curriculum III

In Curriculum II the columns of $X_i$ are global $\ell_0$ minimizers, but their linear combinations in the classes $\mathcal{C}_i$ or the training samples are generally not, which can be fixed by the modification in Figure 4. All blocks individually work as before, but instead of allowing all possible sparse linear combinations of the columns, we only allow one non-zero contribution from each block column. This ensures the sparsity requirements in Remark 5.2 so that all problems in class are global $\ell_0$ minimizers.

**Lemma 5.5.** *For all nodes $i \in \mathcal{I}$ in Curriculum III, define the classes by*

$$\mathcal{C}_i := \{x \in \mathbb{R}^n : x = X_i z, \ z \text{ is } t\text{-sparse}$$

$$z \text{ has at most one non-zero entry per block column of } X_i\}$$

*Then all elements of $\mathcal{C}_i$ are global are global $\ell_0$ minimizers.*

As for Curriculum II Theorem 4.2 is not applicable to show that this tree is learnable. Since the $y$ and $z$ entries are non-negative, this allows us to build a curriculum to learn one arbitrary 1-in-3-SAT problem in a larger class of random signed problems. If we can build an entire curriculum that is fully contained in 1-in-3-SAT itself remains open.

### 5.4 Numerical Experiments

Table 1 contains results for Curricula II and III. All $\ell_1$-minimizations problems are solved by gradient descent in the kernel of $Ax = b$ and the sparse factorization is implemented by $\ell_4$-maximization Zhai et al. (2020). Solutions on the leaf nodes are given instead of brute force solved. As in Welper (2021), Algorithm 1 contains an additional grader that sorts out wrong solutions from SOLVE, which often depend on the gradient descent accuracy. SCALE is implemented by snapping the output of SPARSEFACTOR to the discrete values $\{-1, 0, 1\}$, which allows exact recovery of all nodes $X_i$, without numerical errors. Further details are given in Appendix C.

- *Curriculum II:* We train three tree nodes on two levels. Grader tests to accuracy $10^{-4}$. The results are the average of 5 independent runs.

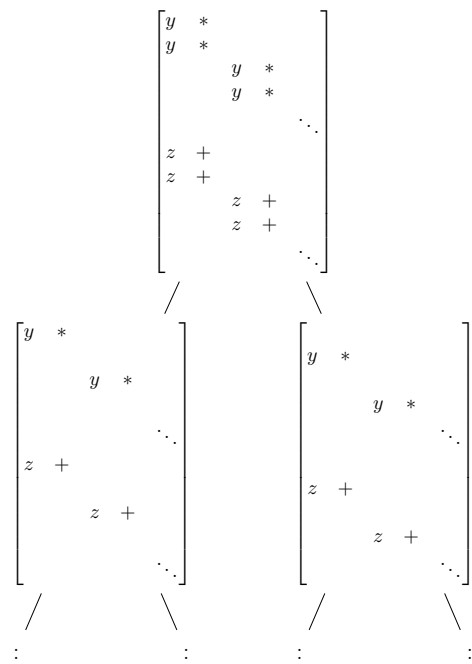

Figure 4: $X_i$ matrices for Curriculum III with $\ell_0$ minimal classes.

|  | Curr. I | | Curr. II |
| --- | --- | --- | --- |
| Depth | 0 | 1 | 0 |
| $m$ | 96 | 96 | 121 |
| $n$ | 128 | 128 | 162 |
| $p\left(X_{\text{child}(i)}\right)$ | 102 | 102 | 459 |
| $\text{Rank}\left(AX_{\text{child}(i)}\right)$ | 96.00 | 62.80 | 113.00 |
| # Samples | 10000 | 10000 | 90000 |
| % VALIDATE | 0.55 | 0.91 | 0.98 |
| $\#(X_{student} = X)$ | 5/5 | 7/10 | 2/2 |

Table 1: Results of numerical experiments, Section 5.4, averaged over all runs and all nodes of given depth. The second but last row shows the percentage of successful training solutions, according to the grader. The last row shows the number of successfully recovered $X_i$ for the given level out of the total number of trials.

- *Curriculum III:* We train one tree node. The training sample matrices (23) are preconditioned per node, not globally as in Theorem 4.2, below. Grader tests to accuracy $10^{-3}$. The results are the average of 2 independent runs.

Table 1 contains the results. It includes average ranks to show that the systems $AX$ are non-trivial with non-zero kernel and the row %VALIDATE shows the percentage of correctly recovered training samples according to the grader. A major bottleneck is the number of training samples for each node, which scales quadratically for $\ell_4$ maximization (but only linear for unique factorization without algorithm Spielman et al. (2012)), up to log factors. The last line shows that in the majority of cases we can recover the tree nodes $X_i$. The misses depend on solver parameters as e.g. iteration numbers and the size of random matrices.

# 6 Conclusion

Although sparse solutions of linear systems are generally hard to compute, many subclasses are tractable. In particular, the prior knowledge $x = Xz$ with sparse $z$ allows us to solve problems with only mild assumptions on $A$. We learn $X$ from a curriculum of easy samples and condensation of knowledge at every tree node. The problems in each class must be compatible so that $AX$ satisfies the null space property. To demonstrate the feasibility of the approach, we show that the algorithms can learn a class $X$ of non-trivial size that contains an arbitrary solution $x$.

The results provide a rigorous mathematical model for some hypothetical principles in human reasoning, including expert knowledge and its training in a curriculum. To be applicable in practice, further research is required, e.g.:

- The mapping of SAT type problems into sparse linear problems lacks several invariances, e.g. a simple reordering of terms may invalidate acquired knowledge. The reduction of SAT or other problems to sparse linear solvers is similar to feature engineering in machine learning.

- For sparse factorization, the required number of samples scales quadratically, up to a log factor, which is the biggest computational bottleneck in the numerical experiments. However, the current implementation uses a standard method and does not use that the parent class $X_i$ can be constructed from its children (12).

- The curriculum is designed so that knowledge can be condensed by sparse factorization, which in itself is a meta-heuristic. One may need to dynamically adapt the condensation heuristic to real data. Since sparse factorization algorithms themselves often rely on $\ell_1$ minimization, similar approaches as discussed in the paper are conceivable.

- Not all relevant knowledge can be combined into one root class $X_0$ so that $AX_0$ satisfies the null space property. Hence, one may need several roots or rather a knowledge graph, together with a decision criterion which node to use for a given problem.

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

# A    Details and Proofs

## A.1    Easy and Hard Problems: Theorems 2.4, 2.5

Theorem 2.4 contains some small changes to the original reference Welper (2021). In the original version (A1) contains two extra inequalities

$$n \geq \bar{c}_1 p \log p, \qquad\qquad \frac{1}{p} \leq \frac{s}{n} \leq \bar{c}_2,$$

which are used to ensure that $X$ has full rank Welper (2021), Proof of Theorem 4.2 with (A3), Item 4. We assume this directly in (A3) and leave out the inequalities.

For Theorem 2.5, the reference Welper (2021) requires the extra assumption that $Ax = b$ has unique $st$ sparse solutions, which is only used to verify that solutions of SOLVE are correct . In our case, this is implicitly contained in (A2), instead.

## A.2    Tree Size: Lemma 3.8

**Lemma A.1** (Lemma 3.8 restated). *Let $s_0$ be the sparsity of the root node of the tree. Assume that each node of the tree has at most $\gamma$ children and that $s_i \bar{t} \gtrsim c s_j t$ for $c \geq 0$ and all $j \in \text{child}(i)$. Then the tree has at most*

$$\gamma^{N+1} = \gamma s_0^{\frac{\log \gamma}{\log(ct/\bar{t})}}$$

*nodes.*

*Proof.* Let $\ell_i$ be the level of a node, i.e. the distance to the root node, and $N$ the maximal level of all nodes. Each level has at most $\gamma^{N-i}$ nodes and thus the full tree has at most

$$\sum_{i=0}^{N} \gamma^{N-i} = \frac{\gamma^{N+1} - 1}{\gamma - 1} \leq \gamma \gamma^N$$

nodes.

It remains to estimate $N$. By induction on the assumption $s_i \bar{t} \geq c s_j t$ we have

$$s_j \leq \left( \frac{\bar{t}}{ct} \right)^{\ell_j} s_0$$

and thus, since necessarily $s_j \geq 1$, we conclude that

$$s_0 \geq \left( \frac{ct}{\bar{t}} \right)^N .$$

Plugging in $\gamma^N = \left( \frac{ct}{\bar{t}} \right)^{N \frac{\log \gamma}{\log ct/\bar{t}}}$ the number of nodes is bounded by

$$\gamma \gamma^N = \gamma \left( \frac{ct}{\bar{t}} \right)^{N \frac{\log \gamma}{\log ct/\bar{t}}} \leq \gamma s_0^{\frac{\log \gamma}{\log ct/\bar{t}}} .$$

$\square$

### A.3 Learnable Trees: Theorem 3.5

**Theorem A.2** (Theorem 3.5 restated)**.** *Let* $\mathcal{C}_i$, $i \in \mathcal{I}$ *be learnable according to Definition 3.4. Then, there exists an implementation of* SPARSEFACTOR *and constants* $c > 0$ *and* $C \geq 0$ *independent of the probability model, dimensions and sparsity, so that with probability at least*

$$1 - C \gamma s_0^{\frac{\log \gamma}{\log(c_s t/\bar{t})}} p^{-c}$$

*the output* $\bar{X}_i = $ TREETRAIN$(\mathcal{C}_i)$ *of Algorithm 2 is a scaled permutation* SCALE$(\bar{X}_i) = $ SCALE$(X_i P)$ *of* $X_i$ *for some permutation matrix* $P$.

*Proof.* The result follows from inductively applying Theorem 2.4 on each node of the tree, starting at its leafs. The assumptions of Theorem 2.4 are easily matched with the given ones, except for (A2), which we verify separately for leaf and non-leaf nodes.

1. *Leave Nodes:* For the leaf nodes (A2) is assumed. This is required because the globally sparsest solution of $Ax = b$ may not be unique, in which case (A2) ensures that we pick an in class solution.

2. *Non-Leave Nodes:* Let $z$ be a column of the training sample $Z$ and $x = X_i z$. By (12), we have

$$x = X_i z = X_{\text{child}(i)} W_{\text{child}(i)} z =: X_{\text{child}(i)} w$$

with $\sqrt{2}t$ sparse $w$ because $W_{\text{child}(i)}$ has $t/\bar{t}$ sparse columns and $z$ is $\sqrt{2}\bar{t}$ sparse, with probability at least $1 - 2p^{-c}$ (see the proof of Theorem 2.4, Item 2, in Welper (2021)). Since $AX_{\text{child}(i)}$ satisfies the $\sqrt{2}t$-RIP, the correct solution $x$ is recovered by the modified $\ell_1$-minimization (4) and hence by SOLVE$_{X_i}$.

Finally, we add up the probabilities. By Theorem 2.4, the probability of failure on each node is at most $Cp^{-c}$. By Lemma 3.8, there are at most $\gamma s_0^{\frac{\log \gamma}{\log(ct/\bar{t})}}$ nodes and thus the result follows from a union bound.

$\square$

*Proof of Corollary 3.6.* By assumption, the student knows the matrices $\bar{X}_i$, which are scaled permutations of $X_i$, i.e. SCALE$(\bar{X}_i) = X_i S P$ for some scaling matrix $S$ and permutation matrix $P$. Since sub-matrices of NSP matrices are also NSP, by Assumption (T5) of learnable trees and removing contributions from siblings,

for all tree nodes $i \in \mathcal{I}$, the scaled product $A[\text{SCALE}(\bar{X}_i)]$ satisfies the null space property of order . Hence, for every problem $x = X_i z = (X_i SP)(P^{-1}S^{-1}z) =: \text{SCALE}(\bar{X}_i)\bar{z}$ with $t$-sparse $z$ in class $\mathcal{C}_i$, the convex $\ell_1$-minimization problem

$$\min_{\bar{z} \in \mathbb{R}^p} \|\bar{z}\|_1 \quad \text{subject to} \quad A\text{SCALE}(\bar{X}_i)\bar{z} = b,$$

recovers $\bar{z}$ and thus the solution $x = \text{SCLAE}(\bar{X}_i)\bar{z}$.

$\square$

## A.4 Split of Global $\ell_0$ Minimizers

This section contains two lemmas that state the splits of $\ell_0$ minimizers are again $\ell_0$ minimizers and that they are linearly independent.

**Lemma A.3** (Lemma 4.3 restated)**.** *Assume the columns of $S \in \mathbb{R}^{n \times q}$ have non-overlapping support and $z \in \mathbb{R}^q$ with non-zero entries. If the vector $x = Sz$ is the solution of the $\ell_0$-minimization problem 14, then the columns of $AS$ are linearly independent.*

*Proof.* Let $x_i$ be the columns of $S$ and assume that the $Ax_i$, $i \in [t]$ are linearly dependent. Then there exists a non-zero $y \in \mathbb{R}^t$ such that $\sum_{i=1}^{t} Ax_i y_i = 0$. Without loss of generality, let $y_1 \neq 0$ so that

$$Ax_1 = -A \sum_{i=2}^{t} x_i \frac{y_i}{y_1}.$$

We use this identity to eliminate $x_1$:

$$b = Ax = A\sum_{i=1}^{t} x_i z_i, = Ax_1 z_1 + A\sum_{i=2}^{t} x_i z_i, = A\sum_{i=2}^{t} x_i z_i \left(1 - \frac{y_i}{y_1} z_0\right) =: A\bar{x}.$$

Since all $x_i$ have disjoint support and all $z_i$ are non-zero, we have $\|\bar{x}\|_0 < \|x\|_0$, which contradicts the assumption that $x$ is a $\ell_0$ minimizer and thus all $Ax_i$, $i \in [n]$ must be linearly independent.

$\square$

**Lemma A.4** (Lemma 4.5 restated)**.** *Assume the columns of $S \in \mathbb{R}^{n \times q}$ have non-overlapping support and $z \in \mathbb{R}^q$ with non-zero entries. If the vector $x = Sz$ is the solution of the $\ell_0$-minimization problem 14, then the columns $S_{\cdot k}$, $k \in [q]$ are global $\ell_0$ optimizers of*

$$S_{\cdot k} \in \min_{x \in \mathbb{R}^n} \|x\|_0 \quad \text{subject to} \quad Ax = AS_{\cdot k}.$$

*Proof.* Assume the statement is wrong. Then for some $k \in [q]$ there is a $y_k$ with

$$\|y_k\|_0 \leq \|S_{\cdot k}\|_0, \quad Ay_k = AS_{\cdot k}.$$

Define

$$\bar{x} := y_k z_k + \sum_{l \neq k} S_{\cdot l} z_l.$$

Then, we have

$$A\bar{x} = Ay_k z_k + A\sum_{l \neq k} S_{\cdot l} z_l. = A\sum_{l} S_{\cdot l} z_l = ASz = Ax$$

and since all $S_{\cdot l}$ have disjoint support and $z_l \neq 0$

$$\|\bar{x}\|_0 = \|y_k\|_0 + \sum_{l \neq k} \|S_{\cdot l}\|_0 < \sum_{l} \|S_{\cdot l}\|_0 = \|x\|_0.$$

This contradicts the assumption that $x$ is a global $\ell_0$ minimiser and hence all $S_{\cdot k}$ must be $\ell_0$ minimizers as well.

$\square$

### A.5 Tree Nodes for Theorem 4.2

This section contains the construction of the matrices $X$ in the tree nodes used in Theorem 4.2.

### A.5.1 Construction of $X$

We follow the idea outlined in Section 4.3. For given matrix $A$ and vector $x$, we construct a decomposition matrix $X \in \mathbb{R}^{n \times p}$ and $z$ so that $x = Xz$ for $t$-sparse $z$ and $AX$ satisfies the null space property. The first condition ensures that $x$ is contained in the class $\mathcal{C}_{<t}$ and the second provides solvers SOLVE. This construction will be used in subsequent sections to define nodes in the curriculum tree. We start with some simple definitions

(M1) By $\mathcal{S}^{m \times n}$ we denote all matrices in $\mathbb{R}^{m \times n}$ whose columns have non-overlapping support.

(M2) $\mathbf{1} := \begin{bmatrix} 1 & \cdots & 1 \end{bmatrix}^T$ with dimensions derived from context.

We split $x$ into $q$ non-overlapping components, which we combine into the columns of a matrix $S \in \mathcal{S}^{n \times q}$ so that $x = S\mathbf{1}$. The matrix $S$ has $q$ columns, which is generally less than the $p$ columns we desire for a rich class given by $X$. A convenient way out is to choose some matrix $Z \in \mathbb{R}^{p \times q}$ with orthonormal columns so that $x = SZ^T Z\mathbf{1} = SZ^T z$ with $z := Z\mathbf{1}$. To ensure sparsity of $z$ and for later tree construction, we confine $Z$ to $\mathcal{S}^{p \times q}$.

(M3) $S \in \mathcal{S}^{n \times q}$ with non-zero columns.

(M4) $Z \in \mathcal{S}^{p \times q}$ with $\ell_2$-normalized columns.

While the matrix $SZ^T$ has the same dimensions as $X$, it is generally low rank and cannot satisfy the NSP. Furthermore, we want a rich class matrix $X$ with further possible random solutions. To this end, we add in a random matrix $R$, but only on blocks of $SZ^T$ that are non-zero to keep sparsity. We define $R$ as follows

(M5) Note that upon permutation of rows and columns $SZ^T$ is a block diagonal matrix with $l \in [q]$ blocks with row indices $\mathrm{supp}(S_{\cdot l})$ and column indices $\mathrm{supp}(Z_{\cdot l})$. If the blocks do not contain all rows and columns, we may enlarge them to some disjoint sets $J_l$ and $K_l$, respectively so that

$$\mathrm{supp}(S_{\cdot l}) \subset J_l, \qquad \mathrm{supp}(Z_{\cdot l}) \subset K_l, \qquad l \in [q].$$

Then each set $J_l \times K_l$ corresponds to one diagonal block that contains one component of $x$ in the columns of $S$. We also use the index free notation

$$\mathcal{J} := \{J_l : l \in [q]\}, \qquad \mathcal{K} := \{K_l : l \in [q]\}, \qquad \mathcal{JK} := \{J_l \times K_l : l \in [q]\},$$

(M6) $R \in \mathbb{R}^{n \times p}$ is block matrix

$$R_{jk} = \begin{cases} \text{i.i.d random} & j, k \in [J, K] \in \mathcal{JK} \\ 0 & \text{else,} \end{cases}$$

whose random entries satisfy

$$\mathbb{E}\left[R_{jk}\right] = 0, \qquad \mathbb{E}\left[R_{jk}^2\right] = 1, \qquad \|R_{jk}\|_{\psi_2} \leq C_\psi$$

for some constant $C_\psi$ and are absolutely continuous with respect to the Lebesgue measure.

Finally, we need a scaling matrix that will be determined below.

(M7) $D \in \mathbb{R}^{n \times n}$ is a diagonal scaling matrix to be determined below.

Then, we define the following class matrix

(M8)
$$X := SZ^T + DR(I - ZZ^T), \tag{25}$$

which is random on the kernel of $Z^T$ and matches the previously constructed $SZ^T$ on the orthogonal complement.

The following lemma summarises several elementary properties of the matrices and vectors in (M1) - (M8) that are used in the proofs below. In particular, they satisfy $x = Xz$ for $z = Z\mathbf{1}$.

**Lemma A.5.** *For the construction (M1) - (M8) we have:*

1. $Z^T Z = I$.

2. $ZZ^T$ *is an orthogonal projector.*

3. *Let* $\operatorname{supp}(Z_{.l}) \subset K \in \mathcal{K}$ *for some column* $l$*. Then*
$$(ZZ^T)_{KL} = \begin{cases} Z_{Kl}Z_{Kl}^T & \text{if } K = L \\ 0 & \text{else.} \end{cases}$$

4. $(ZZ^T)_{KL} = 0$ *for all* $K \neq L \in \mathcal{K}$*.*

5. $(ZZ^T)_{KK}$ *is an orthogonal projector for all* $K \in \mathcal{K}$*.*

6. *For all* $u \in \mathbb{R}^p$ *we have*
$$\sum_{K \in \mathcal{K}} \left\| (ZZ^T)_{KK} u_K \right\|^2 = \left\| Z^T u \right\|^2.$$

7. *For all* $u \in \mathbb{R}^p$ *we have*
$$\sum_{K \in \mathcal{K}} \left\| (I - ZZ^T)_{K.} u \right\|^2 \leq \|u\|^2.$$

8. *For* $z = Z\mathbf{1}$*, we have* $ZZ^T z = z$*.*

9. *For* $x = S\mathbf{1}$ *and* $z = Z\mathbf{1}$*, we have* $SZ^T z = x$*.*

10. *For* $x = S\mathbf{1}$ *and* $z = Z\mathbf{1}$*, we have* $Xz = x$*.*

*Proof.*        1. Since $Z$ is normalized and $Z \in \mathcal{S}^{p \times q}$, all columns are orthonormal.

2. $ZZ^T$ is symmetric and with Item 1 we have $(ZZ^T)(ZZ^T) = Z(Z^T Z)Z^T = ZZ^T$.

3. We have $(ZZ^T)_{KL} = \sum_{l=1}^q (Z_{.l}Z_{.l}^T)_{KL} = \sum_{l=1}^q Z_{Kl}Z_{Ll}^T$, which reduces to the formula in the lemma because $K \neq L$ are disjoint and $\operatorname{supp} Z_{.l} \subset K$.

4. Follows directly from Item 3.

5. Follows directly from Item 3 because the vectors $Z_{Kl}$ is normalized.

6. For every $K \in \mathcal{K}$, let $l \in [q]$ be the corresponding index with $\operatorname{supp}(Z_{.l}) \subset K$. Then, we have

$$\sum_{K \in \mathcal{K}} \left\| (ZZ^T)_{KK} u_K \right\|^2 = \sum_{K,l=1}^q \left\| Z_{Kl}Z_{Kl}^T u_K \right\|^2$$

$$= \sum_{K,l=1}^q (Z_{Kl}^T u_K)^2 = \sum_{l=1}^q (Z_{.l}^T u)^2 = \left\| Z^T u \right\|^2,$$

where in the first equality we have used Item 3, in the second that all $Z_{Kl}$ are normalized and in the third that $\operatorname{supp}(Z_{Kl}) \subset K$.

7. From Item 3, we have

$$(I - ZZ^T)_{K.}u = u_K - \sum_{L \in \mathcal{K}} (ZZ^T)_{KL} u_L = u_K - (ZZ^T)_{KK} u_K.$$

Since by Item 5 the matrix $(I - ZZ^T)_{KK}$ is a projector, it follows that

$$\sum_{K \in \mathcal{K}} \left\| (I - ZZ^T)_{K.}u \right\|^2 = \sum_{K \in \mathcal{K}} \left\| (I - ZZ^T)_{KK} u_K \right\|^2$$

$$\leq \sum_{K \in \mathcal{K}} \left\| (I - ZZ^T)_{KK} \right\|^2 \|u_K\|^2 \leq \|u\|^2.$$

8. With Item 1 we have $ZZ^T z = ZZ^T Z\mathbf{1} = Z\mathbf{1} = z.$

9. With Item 1 we have $SZ^T z = SZ^T Z\mathbf{1} = S\mathbf{1} = x.$

10. Follows directly from the previous items.

$\square$

### A.5.2 Expectation and Concentration

For the proof of RIP and null space properties, we need expectation and concentration results for $\|AXu\|$ for an arbitrary $u$.

**Lemma A.6.** *Let $u \in \mathbb{R}^p$, $A \in \mathbb{R}^{m \times n}$ and $X$ be the matrix defined in* (25). *Then*

$$\mathbb{E}\left[\|AXu\|^2\right] = \left\|ASZ^T u\right\|^2 + \sum_{[J,K] \in \mathcal{JK}} \|AD_{.J}\|_F^2 \left[\|u_K\|^2 - \left\|(ZZ^T)_{KK} u_K\right\|^2\right].$$

*Proof.* Since $R$ is zero outside of the blocks $R_{JK}$ for $[J,K] \in \mathcal{JK}$, we have

$$Xu = [SZ^T + DR(I - ZZ^T)]u = SZ^T u + \sum_{[J,K] \in \mathcal{JK}} D_{.J} R_{JK}(I - ZZ^T)_{K.}u$$

and thus

$$\mathbb{E}\left[\|AXu\|^2\right] = \mathbb{E}\left[\left\|ASZ^T u + \sum_{[J,K] \in \mathcal{JK}} AD_{.J} R_{JK}(I - ZZ^T)_{K.}u\right\|^2\right]$$

$$= \left\|ASZ^T u\right\|^2 + \sum_{[J,K] \in \mathcal{JK}} \left\|AD_{.J} R_{JK}(I - ZZ^T)_{K.}u\right\|^2$$

$$= \left\|ASZ^T u\right\|^2 + \sum_{[J,K] \in \mathcal{JK}} \|AD_{.J}\|_F^2 \left\|(I - ZZ^T)_{K.}u\right\|^2,$$

where in the third line we have used Lemma B.1 and in the second that the cross terms

$$\left\langle ASZ^T u, \ \sum_{[J,K] \in \mathcal{JK}} AD_{.J} \mathbb{E}\left[R_{JK}\right] (I - ZZ^T)_{K.}u \right\rangle = 0$$

and

$$\left\langle AD_{.\bar{J}} \mathbb{E}\left[R_{\bar{J}\bar{K}}\right] (I - ZZ^T)_{\bar{K}.}u, \ AD_{.J} \mathbb{E}\left[R_{JK}\right] (I - ZZ^T)_{K.}u \right\rangle = 0$$

vanish because $\mathbb{E}[R_{JK}] = 0$ and in the last equation we have split the expectation because $R_{JK}$ and $R_{\bar{J}\bar{K}}$ are independent for all cross terms $(\bar{J}, \bar{K}) \neq (J, K)$. We simplify the last term

$$
\begin{aligned}
\left\| (I - ZZ^T)_{K.}u \right\|^2 &= \left\| u_K - \sum_{L \in \mathcal{K}} (ZZ^T)_{KL} u_L \right\|^2 \\
&= \left\| u_K - (ZZ^T)_{KK} u_K \right\|^2 \\
&= \|u_K\|^2 - \left\| (ZZ^T)_{KK} u_K \right\|^2,
\end{aligned}
$$

where the second and third lines follow from Items 4 and 5 in Lemma A.5, respectively. Hence, we obtain

$$
\mathbb{E}\left[ \|AXu\|^2 \right] = \left\| ASZ^T u \right\|^2 + \sum_{[K,J] \in \mathcal{JK}} \|AD_{.K}\|_F^2 \left[ \|u_K\|^2 - \left\| (ZZ^T)_{KK} u_K \right\|^2 \right].
$$

$\square$

If $AS$ has orthonormal columns, we can simplify the expectation. Since this is generally not true, we rename $A \to M$, which will be a preconditioned variant of $A$ later.

**Lemma A.7.** *Let $u \in \mathbb{R}^p$ and $M \in \mathbb{R}^{m \times n}$. With $X$, $S$ and $D$ defined in (25), assume that $MS$ has orthonormal columns and the diagonal scaling is chosen as $D_j = \|M_{.J}\|_F^{-1}$ for all $j$ in block $J \in \mathcal{J}$. Then*

$$
\mathbb{E}\left[ \|MXu\|^2 \right] = \|u\|^2.
$$

*Proof.* The result follows from Lemma A.6 after simplifying several terms. First, since $MS$ has orthonormal columns, we have $(MS)^T(MS) = I$ and thus

$$
\left\| MSZ^T u \right\|^2 = u^T Z (MS)^T (MS) Z^T u = u^T ZZ^T u = \left\| Z^T u \right\|^2.
$$

Second, for arbitrary $j \in J$, by definition of the scaling $D$, we have

$$
\|MD_{.J}\|_F^2 = \|M_{.J}\|_F^2 |D_j|^2 = \|M_{.J}\|_F^2 \|M_{.J}\|_F^{-2} = 1.
$$

Finally, from Lemma A.5 Item 6, we have

$$
\sum_{K \in \mathcal{K}} \left\| (ZZ^T)_{KK} u_K \right\|^2 = \left\| Z^T u \right\|^2.
$$

Plugging into Lemma A.6, we obtain

$$
\begin{aligned}
\mathbb{E}\left[ \|MXu\|^2 \right] &= \left\| MSZ^T u \right\|^2 + \sum_{[J,K] \in \mathcal{JK}} \|MD_{.J}\|_F^2 \left[ \|u_K\|^2 - \left\| (ZZ^T)_{KK} u_K \right\|^2 \right]. \\
&= \left\| Z^T u \right\|^2 + \left( \sum_{[J,K] \in \mathcal{JK}} \|u_K\|^2 \right) - \left\| Z^T u \right\|^2 \\
&= \|u\|^2.
\end{aligned}
$$

$\square$

Next, we prove concentration inequalities for the random matrix $X$.

**Lemma A.8.** *Let $u \in \mathbb{R}^p$ and $M \in \mathbb{R}^{m \times n}$. With $X$, $S$ and $D$ defined in (25), assume that $MS$ has orthonormal columns and the diagonal scaling is chosen as $D_j = \|M_{.J}\|_F^{-1}$ for all $j$ in block $J \in \mathcal{J}$. Then*

$$
\left\| \|MXu\|^2 - \|u\| \right\|_{\psi_2} \leq C C_\psi^2 \max_{J \in \mathcal{J}} \frac{\|M_{.J}\|}{\|M_{.J}\|_F} \|u\|.
$$

*Proof.* The result follows from Lemma B.4 after we have vectorized $R$. To this end, let $\mathrm{vec}(\cdot)$ be the vectorization, which identifies a matrix $\mathbb{R}^{a \times b}$ with a vector in $(\mathbb{R}^a) \otimes (\mathbb{R}^b)'$ for any dimensions $a$, $b$. Then, since for all matrices $ABu = (A \otimes u^T)\,\mathrm{vec}(B)$, we have

$$MD_{.J}R_{JK}(I - (ZZ^T)_K. u = \left[MD_{.J} \otimes u^T(I - (ZZ^T)_{K.}^T \right]\mathrm{vec}\,(R_{JK})$$

so that

$$\begin{aligned}
MXu &= [MSZ^T + MDR(I - ZZ^T)]u \\
&= MSZ^T u + \sum_{[J,K]\in\mathcal{JK}} MD_{.J}R_{JK}(I - ZZ^T)_K. u \\
&= MSZ^T u + \sum_{[J,K]\in\mathcal{JK}} \left[MD_{.J} \otimes u^T(I - ZZ^T)_{K.}^T \right]\mathrm{vec}\,(R_{JK}) \\
&=: \mathcal{B} + \mathcal{AR},
\end{aligned}$$

with the block matrix and vectors

$$\begin{aligned}
\mathcal{A} &:= \left[MD_{.J} \otimes u^T(I - ZZ^T)_{K.}^T\right]_{[J,K]\in\mathcal{JK}} \\
\mathcal{R} &:= \left[\mathrm{vec}\,(R_{JK})\right]_{[J,K]\in\mathcal{JK}} \\
\mathcal{B} &:= MSZ^T u.
\end{aligned}$$

Using Lemma B.2 in the fist equality and Lemma A.7 in the last, we have

$$\|\mathcal{A}\|_F^2 + \|\mathcal{B}\|^2 = \mathbb{E}\left[\|\mathcal{AR} + \mathcal{B}\|^2\right] = \mathbb{E}\left[\|MXu\|^2\right] = \|u\|^2.$$

Furthermore, we have

$$\begin{aligned}
\|\mathcal{A}\| &\leq \left(\sum_{[J,K]\in\mathcal{JK}} \left\|MD_{.J} \otimes u^T(I - ZZ^T)_{K.}^T\right\|^2\right)^{1/2} \\
&= \left(\sum_{[J,K]\in\mathcal{JK}} \|MD_{.J}\|^2 \left\|(I - ZZ^T)_K. u\right\|^2\right)^{1/2} \\
&= \max_{J\in\mathcal{J}} \|MD_{.J}\| \left(\sum_{K\in\mathcal{K}} \left\|(I - ZZ^T)_K. u\right\|^2\right)^{1/2} \\
&\leq \max_{J\in\mathcal{J}} \|MD_{.J}\| \|u\|,
\end{aligned}$$

where in the last inequality we have used Lemma A.5, Item 7. Thus, with Lemma B.4, we have

$$\left\|\|MXu\| - \|u\|\right\|_{\psi_2} = \left\|\|\mathcal{AR} + \mathcal{B}\| - \left(\|\mathcal{A}\|_F^2 + \|\mathcal{B}\|^2\right)^{1/2}\right\|_{\psi_2}$$

$$\leq CC_\psi^2 \|\mathcal{A}\| \leq CC_\psi^2 \max_{J\in\mathcal{J}} \|MD_{.J}\| \|u\|.$$

We can further estimate the right hand side with the definition of diagonal scaling $D$

$$\|MD_{.J}\| = \|M_{.J}D_{JJ}\| = \frac{\|M_{.J}\|}{\|M_{.J}\|_F},$$

which completes the proof.

$\square$

### A.5.3 RIP of $MX$

We do not show the RIP for $AX$ directly, but for a preconditioned variant. Since we determine the preconditioner later, we first state results for a generic matrix $MX$. With the expectation and concentration inequalities from the previous section, the proof of the RIP is standard, see e.g. Baraniuk et al. (2008); Foucart & Rauhut (2013); Kasiviswanathan & Rudelson (2019). We first show a technical lemma.

**Lemma A.9.** *Let $A \in \mathbb{R}^{m \times n}$ and assume that there is a $\frac{\epsilon}{4}$ cover $\mathcal{N} \subset S^{n-1}$ of the unit sphere $S^{n-1}$ with*

$$\left| \|Ax_i\| - 1 \right| \leq \frac{\epsilon}{2} \quad \text{for all } x_i \in \mathcal{N}.$$

*Then*

$$(1 - \epsilon)\|x\| \leq \|Ax\| \leq (1 + \epsilon)\|x\| \quad \text{for all } x \in \mathbb{R}^n.$$

*Proof.* Let $x \in S^{n-1}$ be the maximizer of the norm so that $\|Ax\| = \|A\|$. Then, there is a element $x_i \in \mathcal{N}$ in the cover with $\|x - x_i\| \leq \frac{\epsilon}{4}$ and we obtain the upper bound

$$\|A\| = \|Ax\| \leq \|Ax_i\| + \|A(x - x_i)\| \leq \|Ax_i\| + \|A\| \frac{\epsilon}{4}$$

$$\Rightarrow \left(1 - \frac{\epsilon}{4}\right) \|A\| \leq \|Ax_i\|$$

$$\Rightarrow \|A\| \leq \frac{1 + \epsilon/2}{1 - \epsilon/4} \leq 1 + \epsilon.$$

With the upper bound and the given assumptions, for arbitrary $x \in S^{n-1}$, we estimate the lower bound by

$$\|Ax\| \geq \|Ax_i\| - \|A(x - x_i)\| \geq \|Ax_i\| - (1 + \epsilon)\|x - x_i\|$$

$$\geq \left(1 - \frac{\epsilon}{2}\right) - (1 + \epsilon)\frac{\epsilon}{4} = 1 - \frac{\epsilon}{2} - \frac{\epsilon}{4} - \frac{\epsilon^2}{4} \geq 1 - \epsilon.$$

The bounds extend from the sphere to all $x \in \mathbb{R}^n$ by scaling.

$\square$

For the following RIP result, we add in an isometry $W \in \mathbb{R}^{p \times p'}$, with $\|W\cdot\| = \|\cdot\|$, which allows us to construct tree nodes $X_i$ from its children by (12) below.

**Lemma A.10.** *Let $W \in \mathbb{R}^{p \times p'}$ be an isometry and for $M \in \mathbb{R}^{m \times n}$, with $X$, $S$ and $D$ defined in (25), assume that $MS$ has orthonormal columns and the diagonal scaling is chosen as $D_j = \|M_{.J}\|_F^{-1}$ for all $j$ in block $J \in \mathcal{J}$. If $\min_{J \in \mathcal{J}} \frac{\|M_{.J}\|_F^2}{\|M_{.J}\|^2} \geq \frac{2t C_\psi^4}{c\epsilon^2} \log \frac{12ep}{t\epsilon}$, then with probability at least $1 - 2\exp\left(-\frac{c}{2} \frac{\epsilon^2}{C_\psi^4} \min_{J \in \mathcal{J}} \frac{\|M_{.J}\|_F^2}{\|M_{.J}\|^2}\right)$ the matrix $MXW$ satisfies the RIP*

$$(1 - \epsilon)\|z\| \leq \|MXWz\| \leq (1 + \epsilon)\|z\| \quad \text{for all } z \text{ with } \|z\|_0 \leq t.$$

*Proof.* Fix a support $T \subset [p']$ with $|T| = t$ and let $\Sigma_T \subset \mathbb{R}^{p'}$ be the subspace of all vectors supported on $T$. By standard volumetric estimates Baraniuk et al. (2008); Vershynin (2018) there is a $\frac{\epsilon}{4}$ cover $\mathcal{N}$ of the unit sphere in $\Sigma_T$ of cardinality

$$|\mathcal{N}| \leq \left(\frac{12}{\epsilon}\right)^t.$$

Since $\|Wz_i\| = \|z_i\|$, $z_i \in \mathcal{N}$, by Lemma A.8 and a union bound, we obtain

$$\Pr\left[\exists z_i \in \mathcal{N} : \left|\|MXWz_i\| - 1\right| \geq \epsilon\right] \leq 2\left(\frac{12}{\epsilon}\right)^t \exp\left(-c\frac{\epsilon^2}{C_\psi^4} \min_{J \in \mathcal{J}} \frac{\|M_{.J}\|_F^2}{\|M_{.J}\|^2}\right).$$

Let us assume that the event fails and thus $\left|\|MXWz_i\| - 1\right| \leq \epsilon$ for all $z_i \in \mathcal{N}$. Then, by Lemma A.9, we have

$$(1 - \epsilon)\|z\| \leq \|MXWz\| \leq (1 + \epsilon)\|z\| \quad \text{for all } z \in \Sigma_T.$$

There are $\binom{p}{t} \leq \left(\frac{ep}{t}\right)^t$ supports $T$ of size $t$ and thus, by a union bound we obtain

$$(1 - \epsilon) \|z\| \leq \|MXWz\| \leq (1 + \epsilon) \|z\| \quad \text{for all } z \text{ with } \|z\|_0 \leq t$$

with probability of failure bounded by

$$2 \left(\frac{ep}{t}\right)^t \left(\frac{12}{\epsilon}\right)^t \exp\left(-c \frac{\epsilon^2}{C_\psi^4} \min_{J \in \mathcal{J}} \frac{\|M_{.J}\|_F^2}{\|M_{.J}\|^2}\right)$$

$$= 2 \exp\left(-c \frac{\epsilon^2}{C_\psi^4} \min_{J \in \mathcal{J}} \frac{\|M_{.J}\|_F^2}{\|M_{.J}\|^2} + t \log \frac{12ep}{t\epsilon}\right)$$

$$\leq 2 \exp\left(-\frac{c}{2} \frac{\epsilon^2}{C_\psi^4} \min_{J \in \mathcal{J}} \frac{\|M_{.J}\|_F^2}{\|M_{.J}\|^2}\right)$$

if

$$t \log \frac{12ep}{t\epsilon} \leq \frac{c}{2} \frac{\epsilon^2}{C_\psi^4} \min_{J \in \mathcal{J}} \frac{\|M_{.J}\|_F^2}{\|M_{.J}\|^2} \Leftrightarrow \min_{J \in \mathcal{J}} \frac{\|M_{.J}\|_F^2}{\|M_{.J}\|^2} \geq \frac{2t C_\psi^4}{c\epsilon^2} \log \frac{12ep}{t\epsilon}.$$

$\square$

### A.5.4 Null Space Property of $AX$

The matrix $MS$ in the RIP results must have orthonormal columns, which is not generally true for $M = A$. However, this is true with a suitable preconditioner that we construct next. The null space property is invariant under preconditioning, which allows us to eliminate it, later.

**Lemma A.11.** *Let $M \in \mathbb{R}^{m \times q}$ with $m \geq q$ have full column rank. Then there is a matrix $T \in \mathbb{R}^{m \times m}$ with condition number $\kappa(T) = \kappa(M)$ such that $TM$ has orthonormal columns.*

*Proof.* Let $M = U\Sigma V^T$ be the singular value decomposition of $M$. Define

$$T := DU^T, \qquad\qquad\qquad D^{-1} := \text{diag}[\sigma_1, \ldots, \sigma_q, \sigma, \ldots, \sigma]$$

for $q \leq m$ singular values $\sigma_i$ and remaining $m - q$ values $\sigma$ in the interval $[\sigma_1, \ldots, \sigma_q]$. Then, we have

$$M^T T^T T M = (V\Sigma^T U^T)(UD^T)(DU^T)(U\Sigma V^T) = V\Sigma^T D^T D\Sigma V^T = VV^T = I,$$

where we have used that $\Sigma^T D^T D\Sigma = I$. By construction, $T$ has singular values $\sigma_1, \ldots, \sigma_q$ and one extra value $\sigma$ bounded by the former so that

$$\kappa(T) = \frac{\sigma_1}{\sigma_q} = \kappa(M).$$

$\square$

**Lemma A.12.** *Let $A \in \mathbb{R}^{m \times n}$ and $T \in \mathbb{R}^{m \times m}$ be invertible. Then*

$$\frac{\|A\|_F}{\|A\|} \leq \kappa(T) \frac{\|TA\|_F}{\|TA\|}.$$

*Proof.* We first show that

$$\|TA\|_F \geq \|T^{-1}\|^{-1} \|A\|_F.$$

Indeed $\|x\| \leq \|T^{-1}\| \|Tx\|$ implies $\|Tx\| \geq \|T^{-1}\|^{-1} \|x\|$ and thus applied to the columns $a_j$ of $A$, we have

$$\|TA\|_F^2 = \sum_{j=1}^n \|Ta_j\|^2 \geq \sum_{j=1}^n \|T^{-1}\|^{-2} \|a_j\|^2 = \|T^{-1}\|^{-2} \|A\|_F^2.$$

With this estimate, we obtain

$$\kappa(T)\frac{\|TA\|_F}{\|TA\|} \geq \|T\|\,\|T^{-1}\|\,\frac{\|T^{-1}\|^{-1}\|A\|_F}{\|T\|\,\|A\|} = \frac{\|A\|_F}{\|A\|}.$$

$\square$

**Corollary A.13.** *Let* $W \in \mathbb{R}^{p \times p'}$ *be an isometry and for* $X$, $S$ *and* $D$ *defined in* (25), *assume that* $AS$ *has full column rank and* $\min_{J \in \mathcal{J}} \frac{\|A_{.J}\|_F^2}{\|A\|_{.J}^2} \geq \frac{2tC_\psi^4}{c\epsilon^2}\kappa(AS)\log\frac{12ep}{t\epsilon}$ . *Then there is an invertible matrix* $T \in \mathbb{R}^{m \times m}$ *so that with the diagonal scaling* $D_j = \|TA_{.J}\|_F^{-1}$ *for all* $j$ *in block* $J \in \mathcal{J}$ *with probability at least* $1 - 2\exp\left(-\frac{c}{2}\frac{\epsilon^2}{C_\psi^4}\frac{1}{\kappa(AS)}\min_{J \in \mathcal{J}}\frac{\|A_{.J}\|_F^2}{\|A_{.J}\|^2}\right)$ *the matrix* $TAXW$ *satisfies the RIP*

$$(1-\epsilon)\|z\| \leq \|TAXWz\| \leq (1+\epsilon)\|z\| \quad \text{for all } z \text{ with } \|z\|_0 \leq t.$$

*Proof.* Since the matrix $AS$ has full column rank by Lemmas A.11 and A.12, there is an invertible matrix $T$ such that

$$\kappa(T) = \kappa(AS), \qquad\qquad TAS \text{ has orthogonal columns}$$
$$\frac{\|A_{.J}\|_F}{\|A_{.J}\|} \leq \kappa(T)\frac{\|TA_{.J}\|_F}{\|TA_{.J}\|} \qquad\qquad \text{for all } J \in \mathcal{J}.$$

Thus, the corollary follows from Lemma A.10 with $M = TA$.

$\square$

The last corollary allows us to recover $x = S\mathbf{1}$ by $\ell_1$-minimization

$$\min_{x \in \mathbb{R}^n} \|x\|_1 \quad \text{subject to} \quad TAx = b,$$

preconditioned by some matrix $T$. This problem is not yet solvable by the student, who generally has no access to the matrix $T$, which is only used by the teacher for the construction of $X$. However, the matrix $T$ is unnecessary for $\ell_1$ recovery because the RIP implies the null space property, which is sufficient for recovery and independent of left preconditioning.

**Corollary A.14.** *Let* $W \in \mathbb{R}^{p \times p'}$ *be an isometry and for* $X$, $S$ *and* $D$ *defined in* (25), *assume that* $AS$ *has full column rank and* $\min_{J \in \mathcal{J}} \frac{\|A_{.J}\|_F^2}{\|A_{.J}\|^2} \geq \frac{2tC_\psi^4}{c\epsilon^2}\kappa(AS)\log\frac{12ep}{t\epsilon}$ . *Then there is an invertible matrix* $T \in \mathbb{R}^{m \times m}$ *so that with the diagonal scaling* $D_j = \|TA_{.J}\|_F^{-1}$ *for all* $j$ *in block* $J \in \mathcal{J}$ *with probability at least* $1 - 2\exp\left(-\frac{c}{2}\frac{\epsilon^2}{C_\psi^4}\frac{1}{\kappa(AS)}\min_{J \in \mathcal{J}}\frac{\|A_{.J}\|_F^2}{\|A_{.J}\|^2}\right)$ *the matrix* $AXW$ *satisfies the null space property of order* $t$

$$\|z_T\|_1 < \|z_{\bar{T}}\|_1 \qquad\qquad \text{for all } z \in \ker(AXW) \text{ and } T \subset [p], |T| \leq t.$$

*with complement* $\bar{T}$ *of* $T$.

*Proof.* Setting $\epsilon = \frac{1}{3}$, changing $t \to 2t$ and adjusting the constants accordingly, with the given conditions and probabilities, the matrix $TAX$ satisfies the $(2t, \frac{1}{3})$-RIP. Thus, by Foucart & Rauhut (2013), proof of Theorem 6.9, $TAX$ satisfies

$$\|z_T\|_1 < \frac{1}{2}\|z\|_1 \qquad\qquad \text{for all } z \in \ker(TAX) \text{ and } T \subset [p], |T| \leq t.$$

This directly implies the null space property of order $t$

$$\|z_T\|_1 < \|z_{\bar{T}}\|_1 \qquad\qquad \text{for all } z \in \ker(TAX) \text{ and } T \subset [p], |T| \leq t.$$

Since $T$ is invertible, $\ker(TAX) = \ker(AX)$, so that also $AX$ satisfies the null space property.

$\square$

**Remark A.15.** *For Corollaries A.13 and A.14, we are particularly interested in applications where $x = S\mathbf{1}$ is the global $\ell_0$-minimizer of $Ax = b$ in 14. Then the full column rank condition of $AS$ is automatically satisfied by Lemma A.3.*

### A.6  Model Tree: Theorem 4.2

Recall that $\kappa(\cdot)$ denotes the condition number.

**Theorem A.16** (Theorem 4.2 restated)**.** *Let $A \in \mathbb{R}^{m \times n}$ and split $x \in \mathbb{R}^n$ into $q = 2^L$, $L \geq 1$ components $S$ given by* (15)*. If*

1. *$AS$ has full column rank.*

2. *On each tree node, we have implementations of $\textsc{Scale}$.*

3. *$\textsc{SolveL}$ satisfies Assumption (A2) on the leaf nodes.*

4.
$$t \gtrsim \log p^2 + \log^3 p, \qquad\qquad 1 \lesssim t \lesssim \sqrt{p} \qquad\qquad (26)$$

5.
$$\min_{J \in \mathcal{J}} \frac{\|A_{\cdot J}\|_F^2}{\|A_{\cdot J}\|^2} \gtrsim t\kappa(AS)L + t\kappa(AS)\log\frac{cqp}{t} \qquad\qquad (27)$$

*for some generic constant c, with probability at least*

$$1 - 2\exp\left(-c\frac{1}{\kappa(AS)}\min_{J \in \mathcal{J}}\frac{\|A_{\cdot J}\|_F^2}{\|A_{\cdot J}\|^2}\right)$$

*there is a learnable binary tree of problem classes $\mathcal{C}_i$, $i \in \mathcal{I}$ of depth $L$, given by matrices $X_i$ and sparsity $t$ so that*

1. *The root class $\mathcal{C}_0$ contains $x$.*

2. *The parents are constructed from the children $X_i = X_{\mathrm{child}(i)}W_{\mathrm{child}(i)}$, where $W_{\mathrm{child}(i)}$ has $t/\bar{t} = 2$ sparse columns.*

3. *The columns of the leaf nodes' $X_i$ are $|J|$ sparse.*

4. *Each class' matrix $X_i$ contains $p$ columns, consisting of columns of $S$, i.e. pieces of $x$, in the leafs and sums thereof in the interior nodes. All other entries are random (dependent between classes) or zero.*

*Proof.* We build a matrix $X$ according to (M1) - (M8) and use the extra matrix $W$ in Corollary A.14 to build a tree out of it. In the following, we denote by $\bar{p}$ the number of columns in $X$ and by $p$ the number of columns in the class matrices $X_i$ that we are going to construct. By assumption, the support of $x$ is partitioned into patches $\{J_1, \ldots, J_q\} = \mathcal{J}$ for which we define a corresponding partition $\mathcal{K} = \{K_1, \ldots, K_q\}$ of $[\bar{p}]$ with all $K_i$ of equal size and $Z$ by

$$Z_{kl} := \begin{cases} 1 & k = k_l \\ 0 & \text{else} \end{cases}$$

for some choices $k_l \in K_l$. The index sets $\mathcal{J}$ and $\mathcal{K}$ are naturally combined by their indices to obtain the pairs $\mathcal{JK}$. With these choices, the matrix $X$ is given by (M1) - (M8).

$X$ is non-zero only on blocks $[J, K] \in \mathcal{JK}$, which allows us to build a tree, whose nodes we index by $i$ in a suitable index set $\mathcal{I}$ with leaf nodes $i \in [q]$. Each node $i$ is associated with a subset $K_i \subset [q]$ that is a union of two children $K_i = \bigcup_{j \in \mathrm{child}(i)} K_j$, starting with leaf nodes $K_i \in \mathcal{K}$, $i \in [q]$, e.g.

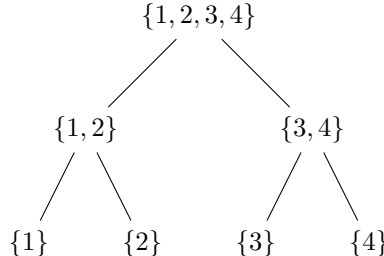

We now define matrices $X_i$ on each node, starting with the leafs

$$X_i := X_{\cdot K_i}$$

for leaf $i$ and then inductively by joining the two child matrices

$$X_i := \begin{bmatrix} X_{j_1} & X_{j_2} \end{bmatrix} \bar{W}_i, \qquad\qquad \bar{W}_i = \frac{1}{\sqrt{2}} \begin{bmatrix} I_{K_{j_1}, K_{j_1}} \\ I_{K_{j_2}, K_{j_2}} \end{bmatrix}$$

for child$(i) = \{j_1, j_2\}$ and identity matrix $I_{\cdot,\cdot}$ on the given index sets. It is easy to join all $\bar{W}_i$ matrices leading up to node $i$ into a single isometry $W_i$ so that

$$X_i = \begin{bmatrix} X_1 & \cdots & X_q \end{bmatrix} W_i.$$

which implies

$$X_{\text{child}(i)} = \begin{bmatrix} X_{j_1} & X_{j_2} \end{bmatrix} = \begin{bmatrix} X_1 & \cdots & X_q \end{bmatrix} W_{\text{child}(i)}, \qquad\qquad W_{\text{child}(i)} = \begin{bmatrix} W_{j_1} & W_{j_2} \end{bmatrix},$$

where again $W_{\text{child}(i)}$ is an isometry because the columns of $W_{j_1}$ and $W_{j_2}$ have non-overlapping support. By Lemma 3.8 the tree has at most $2^{L+1}$ nodes and thus, if

$$\min_{J \in \mathcal{J}} \frac{\|A_{\cdot J}\|_F^2}{\|A_{\cdot J}\|^2} \geq \frac{2t C_\psi^4}{c \epsilon^2} \kappa(AS) \log \frac{12 e \bar{p}}{t \epsilon} \tag{28}$$

by Corollary A.14 and union bound over all tree nodes, with probability at least

$$1 - 42^L \exp\left( -\frac{c}{2} \frac{\epsilon^2}{C_\psi^4} \frac{1}{\kappa(AS)} \min_{J \in \mathcal{J}} \frac{\|A_{\cdot J}\|_F^2}{\|A_{\cdot J}\|^2} \right)$$

all nodes $X_{\text{child}(i)}$ satisfy the $t$-NSP. For this probability to be close to one, $\log 2^L$ must be smaller than say half the exponent

$$L \gtrsim \log 2^L \leq -\frac{c}{4} \frac{\epsilon^2}{C_\psi^4} \frac{1}{\kappa(AS)} \min_{J \in \mathcal{J}} \frac{\|A_{\cdot J}\|_F^2}{\|A_{\cdot J}\|^2} \qquad \Leftrightarrow \qquad \min_{J \in \mathcal{J}} \frac{\|A_{\cdot J}\|_F^2}{\|A_{\cdot J}\|^2} \gtrsim \frac{t C_\psi^4}{\epsilon^2} \kappa(AS) \log s.$$

Combining this with the NSP condition (28), if

$$\min_{J \in \mathcal{J}} \frac{\|A_{\cdot J}\|_F^2}{\|A_{\cdot J}\|^2} \gtrsim \frac{t C_\psi^4}{\epsilon^2} \kappa(AS) L + \frac{t C_\psi^4}{\epsilon^2} \kappa(AS) \log \frac{12 e \bar{p}}{t \epsilon},$$

with probability at least

$$1 - 2 \exp\left( -\frac{c}{2} \frac{\epsilon^2}{C_\psi^4} \frac{1}{\kappa(AS)} \min_{J \in \mathcal{J}} \frac{\|A_{\cdot J}\|_F^2}{\|A_{\cdot J}\|^2} \right)$$

all nodes $X_{\text{child}(i)}$ satisfy the $t$-NSP. This yields the statements in the proposition if we choose $\epsilon \sim 1$ and $C_\psi \sim 1$, without loss of generality.

Let us verify the remaining properties of learnable trees. By construction, we have $t/\bar{t} = 2$ and $\gamma = 2$ and $\bar{p} = qp$. Since all random samples in $X$ are absolutely continuous with respect to the Lebesgue measure, the probability of rank deficit $X_i$ is zero. The remaining assumptions are given, with the exception of the first two inequalities in (A1). Renaming the number of training samples $q$, whose name is already used otherwise here, to $r$, they state that $t \geq c \log r$ and $r > cp^2 \log^2 p$ and thus imply that $t \geq \log p^2 + \log^3 p$, which is sufficient since the number of training samples $r$ is at the disposal of the teacher.

$\square$

## B   Technical Supplements

**Lemma B.1.** *Let $R \in \mathbb{R}^{n \times p}$ be a i.i.d. random matrix with mean zero entries of variance one. Then for any $A \in \mathbb{R}^{m \times n}$ and $u \in \mathbb{R}^p$ we have*

$$\mathbb{E}\left[\|ARu\|^2\right] = \|A\|_F^2 \|u\|^2.$$

*Proof.* Since $\mathbb{E}[R_{ik}R_{jl}] = \delta_{ij}\delta_{kl}$, we have

$$\begin{aligned}
\mathbb{E}\left[\|ARu\|^2\right] &= \mathbb{E}\left[\langle ARu, ARu \rangle\right] \\
&= \mathbb{E}\left[\sum_{ijkl} u_k R_{ik} (A^T A)_{ij} R_{jl} u_l\right] \\
&= \sum_{ijkl} (A^T A)_{ij} u_k u_l \mathbb{E}\left[R_{ik} R_{jl}\right] \\
&= \sum_{ik} (A^T A)_{ii} u_k u_k \\
&= \|A\|_F^2 \|u\|^2.
\end{aligned}$$

$\square$

**Lemma B.2.** *Let $A \in \mathbb{R}^{m \times n}$ be a matrix, $b \in \mathbb{R}^m$ be a vector and $x \in \mathbb{R}^n$ a i.i.d. random vector with $\mathbb{E}[x_j] = 0$, $\mathbb{E}[x_j^2] = 1$. Then*

$$\mathbb{E}\left[\|Ax + b\|^2\right] = \|A\|_F^2 + \|b\|^2.$$

*Proof.* Since $b$ is not random, we have

$$\mathbb{E}\left[\|Ax + b\|^2\right] = \mathbb{E}\left[\|Ax\|^2\right] + \|b\|^2 = \|A\|_F^2 + \|b\|^2,$$

where in the last equality we have used Lemma B.1 with $\mathbb{R}^{n \times 1}$ matrix $R = x$ and $u = [1] \in \mathbb{R}^1$.

$\square$

The following result is a slight variation of Vershynin (2018), Theorem 6.3.2.

**Lemma B.3.** *Let $A \in \mathbb{R}^{m \times n}$ be a matrix, $b \in \mathbb{R}^m$ be a vector and $x \in \mathbb{R}^n$ a i.i.d. random vector with $\mathbb{E}[x_j] = 0$, $\mathbb{E}[x_j^2] = 1$ and $\|x\|_{\psi_2} \leq C_\psi$. Then*

$$\Pr\left[\left|\|Ax + b\|^2 - \|A\|_F^2 - \|b\|^2\right| \geq \epsilon \left(\|A\|_F^2 + \|b\|^2\right)\right]$$

$$\leq 8 \exp\left[-c \min(\epsilon^2, \epsilon) \frac{\|A\|_F^2 + \|b\|^2}{C_\psi^4 \|A\|^2}\right].$$

*Proof.* We decompose

$$\|Ax + b\|^2 - \|A\|_F^2 - \|b\|^2 = \|Ax\|^2 + 2\langle Ax, b\rangle + \|b\|^2 - \|A\|_F^2 - \|b\|^2$$
$$= \left(\|Ax\|^2 - \|A\|_F^2\right) + 2\langle Ax, b\rangle$$

so that

$$\Pr\left[\pm\left(\|Ax + b\|^2 - \|A\|_F^2 - \|b\|^2\right) \geq \epsilon\left(\|A\|_F^2 + \|b\|^2\right)\right]$$
$$\leq \Pr\left[\pm\left(\|Ax\|^2 - \|A\|_F^2\right) \pm 2\langle Ax, b\rangle \geq \epsilon\left(\|A\|_F^2 + \|b\|^2\right)\right]$$
$$\leq \Pr\left[\pm\left(\|Ax\|^2 - \|A\|_F^2\right) \geq \epsilon\|A\|_F^2\right] + \Pr\left[\pm 2\langle Ax, b\rangle \geq \epsilon\|b\|^2\right].$$

It remains to estimate the two probabilities on the right hand side. Since $\mathbb{E}\left[x_j^2\right] = 1$, we have $C_\psi \gtrsim 1$ and thus from the proof of Theorem 6.3.2 in Vershynin (2018), we have

$$\Pr\left[\pm\left(\|Ax\|^2 - \|A\|_F^2\right) \geq \epsilon\|A\|_F^2\right] \leq 2\exp\left[-c\min(\epsilon^2, \epsilon)\frac{\|A\|_F^2}{C_\psi^4\|A\|^2}\right]$$

and from Hoeffding's inequality, we have

$$\Pr\left[\pm 2\langle Ax, b\rangle \geq \epsilon\|b\|^2\right] \leq 2\exp\left[-c\epsilon^2\frac{\|b\|^4}{C_\psi^2\|A^T b\|^2}\right] \leq 2\exp\left[-c\epsilon^2\frac{\|b\|^2}{C_\psi^4\|A^T\|^2}\right].$$

$\square$

The following result is a slight variation of Vershynin (2018), Theorem 6.3.2.

**Lemma B.4.** *Let $A \in \mathbb{R}^{m \times n}$ be a matrix, $b \in \mathbb{R}^m$ be a vector and $x \in \mathbb{R}^n$ a i.i.d. random vector with $\mathbb{E}[x_j] = 0$, $\mathbb{E}\left[x_j^2\right] = 1$ and $\|x\|_{\psi_2} \leq C_\psi$. Then*

$$\left\|\|Ax + b\| - \left(\|A\|_F^2 + \|b\|^2\right)^{1/2}\right\|_{\psi_2} \leq CC_\psi^2\|A\|$$

*for some constant $C \geq 0$.*

*Proof.* We use a standard argument, e.g. from the proof of Theorem 6.3.2 in Vershynin (2018). An elementary computation shows that for $\delta^2 = \min(\epsilon^2, \epsilon)$ and any $a, b \in \mathbb{R}$, we have

$$|a - b| \geq \delta b, \quad \Rightarrow \quad |a^2 - b^2| \geq \epsilon b^2.$$

With $a = \|Ax + b\|$ and $b = \left(\|A\|_F^2 + \|b\|^2\right)^{1/2}$ and Lemma B.3, this implies

$$\Pr\left[\left|\|Ax + b\| - \left(\|A\|_F^2 - \|b\|^2\right)^{1/2}\right| \geq \delta\left(\|A\|_F^2 + \|b\|^2\right)^{1/2}\right]$$
$$\leq 8\exp\left[-c\delta^2\frac{\|A\|_F^2 + \|b\|^2}{C_\psi^4\|A\|^2}\right].$$

This shows Subgaussian concentration and thus the $\psi_2$-norm of the lemma.

$\square$

## C  Implementation Details

### Details for Curriculum I in Section 5.3.1

*Proof of Lemma 5.3.* The curriculum satisfies (M1) – (M8) with the index sets

$$\Big[\underbrace{1,\ldots,|J|}_{J_1},\quad \ldots \quad,\underbrace{n-|J|,\ldots,n}_{J_q}\Big], \qquad\qquad \Big[\underbrace{1,\ldots,|K|}_{K_1},\quad \ldots \quad,\underbrace{p-|K|,\ldots,p}_{K_q}\Big]$$

and $Z = \begin{bmatrix} e_1 & e_{|K|+1} & e_{2|K|+1} & \ldots \end{bmatrix}$ with unit basis vectors $e_k$ for the first index in each block $K_i$. Hence, it is a special case of the construction in the proof of Theorem 4.2 and all conclusions of the theorem are applicable. $\qquad\square$

### Details for the implementation in Section 5.4:

1. The teacher provides a left preconditioned matrix $TA$ in every tree node. This allows RIP instead of weaker NSP conditions, as in Corollary A.13 versus Corollary A.14. For Curriculum II $T$ is uniform for all tree nodes, for Curriculum III, it is computed individually for each node.

2. Unlike (21) in the split $X := SZ^T + DR(I - ZZ^T)$ between deterministic and random part, we use no balancing $D$ in the experiments.

3. As a result, all tree node $X_i$ have entries in $\{-1, 0, 1\}$ so that we implement SCALE by snapping to these discrete values.

## D  Glossary

### Algorithms

| | |
|---|---|
| SOLVE$(A, b)$ | $\ell_0$ minimizer for easy problems, Section 2.2. |
| SPARSEFACTOR$(Y)$ | Sparse matrix factorization $Y = XZ$, Section 2.2. |
| SCALE | Rescaling after matrix factorization, Section 2.2, Definition 3.4. |
| TRAIN$(A, b_1, \ldots, b_q)$ | Find class $X$ from samples, Algorithm 1. |
| SOLVEL | $\ell_0$ minimizer for easy problems in leaf nodes, Definition 3.4. |
| TREETRAIN$(\mathcal{C}_i)$ | Find $X_i$ for all tree nodes $i$ from samples, Algorithm 2. |

### Dimensions

$A \in \mathbb{R}^{m \times n}$
$X \in \mathbb{R}^{n \times p}$
$Z \in \mathbb{R}^{p \times q}$

### Sparsities

| | |
|---|---|
| $s$ | Sparsity of the columns of $X$. |
| $t$ | (Expected) Sparsity of the columns of $Z$ for problems class $\mathcal{C}$. |
| $\bar{t}$ | (Expected) Sparsity of the columns of $Z$ for easy problems in class $\mathcal{C}_{\text{easy}} \subset \mathcal{C}$. |

### Tree

| | |
|---|---|
| $\mathcal{I}$ | Indices of tree nodes, Section 3.1. |
| child$(i)$ | Children of node $i$, Section 3.1. |
| $W_{\text{child}(i)}$ | (13). |
| $X_{\text{child}(i)}$ | (13). |

