# OpenReview forum: "Learning Trees of $\ell_0$-Minimization Problems"
_TMLR — Rejected by TMLR_

### Review · Reviewer_N1a4 · 2023-07-03

**Summary Of Contributions:**

The authors study the classical problem of sparse recovery: $\min ||x||_0: Ax=b$.

Sparse recovery (SR) is an important NP-hard problem that appears in any settings in practice and has a great deal of research into tractable subclasses such as matrices A satisfying the restricted isometry property and null-space properties. In this work, the authors take a new angle on tractable subclasses for SR by considering constrained solution spaces, in particular under the assumption that the solution x is of the form $x=Xz$, i.e. a linear combination of the columns of some pre-determined "knowledge matrix" X.

The main contribution of the authors is the introduction of a new model to learn the knowledge matrix X through a hierarchical "curriculum" aimed to capture how humans learn to solve hard problems through the acquirement of expert knowledge. The basic idea is to divide into a tree of easier (less sparse) sub-problems C_i with constraints X_i such that each X_i is a linear sum over its children, (including the root X). The authors argue that if the X_i are sufficiently random and the learner is given "random" questions from the easier subclasses (akin to say homework problems), it is possible to solve the questions and reverse engineer the constraints X_i, which are then used to construct X itself. Once X is learned, the authors give conditions under which AX satisfies the nullspace condition and one can hence solve $\min ||z||_0: AXz=b$.

Finally, the authors prove that such curriculum can indeed be constructed for any fixed solution x under weak constraints on the matrix A. They then look at a set of problems inspired by the classical NP-hard 1-in-3-SAT, giving several example curricula for fixed instances. Finally the authors run some heuristic numerical experiments to validate that their model can be implemented practically at a small scale.


**Audience:**

Yes

**Claims And Evidence:**

Yes

**Requested Changes:**

*In order to recommend acceptance, the following issues must be addressed:*

The authors need to formally develop the mathematical framework that they are working in, and state their results within formal theorem statements in the model and not as discussion and remarks in free-form exposition. It is often unclear what are assumptions and what are part of the general mathematical framework, and what problem is being solved. For instance, unless I am misunderstanding, it does not seem to be the case that the authors prove any new instances are actually tractable within the standard model of computation, whereas one might get this impression from reading the introduction. This is fine, but the entire main body needs to be re-written taking this into account and clarifying what is actually proved. There are many works in learning theory on "machine teaching" and models of computation that use teachers (see e.g. works on teaching dimension, recursive teaching dimension, etc...) . Please include reference to such material and discuss how the model in this paper relates to prior work in the learning literature.

On a related note, the first sentence of the paper seems misleading re: 3-SAT. As far as I can tell "efficiently solvable subclasses of 3-SAT" are not considered in this paper.

Remark 3.1 is a good example of something that should appear in a more formal theorem statement somewhere. Do not include extra assumptions you need for results in a separate remark after the fact (unless it is clear it is extraneous to the main theorem statement and an afterthought). Please write a formal theorem statement with your assumptions and what you prove in the formal model of learning you introduce.

In Section 4, what is the implication that the curriculum I satisfies (M1)-(M8)? Please give formal statements of what the guarantees are.

In Section 5.2, why does considering a "larger" class of A help? Clarify what is actually being proved in this section, again there should be a theorem statement not just exposition — what part of the family is random in "mostly random signed problems?" Isn’t it the curricula that are random, not the problems?

At the bottom of page 4, it is mentioned that “more structure is required to ensure x is indeed the \ell_0 optimizer.” The authors need to spend substantially more time justifying this point. Whether or not the particular methods laid out are indeed solving for the optimizer should be made clear formally in each statement, and the assumptions needed for this should be laid out clearly in theorem statements and in the proof. As the paper stands, it is never really clear when this holds. Does Remark 5.2 imply that in the 1-in-3 SAT setup Xz is always the true ell_0 minimizer for any sparse solution z? This should be stated within a formal theorem. Same for all parts in the main body.

In (M5) should supp(X.,l) be SZ^T? In general I found (M5) confusing, and much of the notation seems to just be undefined (e.g. what is K_I?). Please use { } or ( ), not [ ] for tuples.

Other missing definitions: "T-sparse matrix", nu in Def 2.2, epsilon in (A4), Xchild(i) and Wchild(i), all parameters in Def 3.5, K In Lemma A.16 (Is this really a partition? Covers all of \bar{p}?), “pre-conditioner T” in the main body (Should this appear in any of the formal theorem statements?), “ I_{K_{j_i},K_{j_i}}.”

In (A1) C_easy is defined by pairs, whereas before it was defined by a constrained solution-space. Please formalize your treatment of these notions and stick to a consistent notation.

The probabilistic quantifier on 4.2 seems wrong as stated. Is it really that the curriculum exists whp, or that whp over X you get the desired curriculum?

Did A get dropped at the top of page 28? Why do the cross terms in the first equality die?

*Below are several less serious issues:*

1. Typos: “notable -> notably”, “heads on -> head on”, “N \neq NP,” “have build up -> have built up”,“form -> from”, “less columns -> fewer columns”, “ca ≤ b ≤ Cb -> ca ≤ b ≤ Ca”, “exits -> exists”, “is is”, “an can”, “leave -> leaf”, “If this can be avoided” -> “Whether this can be avoided”,
2. Top of page 24: should w be 2t sparse?,
3. Should “AXchild(i) satisfies the √ 2t-RIP” be just with high probability?
4. Top of page 31, what is tau? Should be eps?

5. Add a figure diagram for the image on the second page.

6. Page 8: says C_easy is defined as before, but then  just uses sparsity, while (A1) relied on being columns of a subgaussian?

7. Is there supposed to be a "related work" indication between pages 2 and 3?

8. The authors spend a great deal of time motivating their model by human learning, but the majority of the paper relies on random constructions within the curriculum which does not particularly match the motivation. Either cut down on the emphasis on the human side and focus on why the model is of mathematical interest, or add more justification for this difference.

**Strengths And Weaknesses:**

The paper’s main strength is in the introduction a new type of mathematical model for sparse recovery that mimics some structures used in human learning and pedagogy. Sparse recovery is an interesting and classical problem, and the new model may be of interest to some portion of TMLR readership. The formal math I was able to check seemed sound (see below for questions on statements I was not able to check).

The paper’s main weakness is in its writing and lack of mathematical clarity: the main body in particular fails to formally define many relevant quantities and notions, and at times it is hard to follow what exactly is being claimed or proven at all. In general, the mathematical model and theorems need to be more strictly formalized and clearly stated in every section for the paper to be considered in publishable form for a theoretical work.

EDIT: The author has largely addressed the above issues, and I have changed my recommendation accordingly.

---

> ### Author Response · Authors · 2023-07-31
> **Reply to Reviewer**
>
> Thank you for the detailed review. I think it has considerably improved the paper. Changes in the manuscript are in blue. Comments below are numbered by the paragraphs in the section "Requested Changes".
>
> 1. *Formal mathematical framework:* I disagree with the reviewer that the paper lacks formality and has hidden extra assumptions. The main results are contained in formal statements in Theorem 3.5 and Theorem 4.2  (new version, renamed for another reviewer) and they reference all required assumptions. In particular Remark 3.1, given by the reviewer as an example for a hidden assumption, was referenced as Def 3.5, bulled point 4 in the original version, although, I see that this was easy to miss.
>
>     That said, the reviewer provides several valid points for improvement. I have changed/added the following:
>
>     - Added an overview of the main theorems to the introduction, in section "New Contributions".
>
>     - Definition 3.5 (original version) of leanable trees was a collection of assumptions that were motivated in the preceeding section and easy to miss (including Remrak 3.1 mentioned by the reviewer). The definition has been split into (T1) - (T8) and embedded into the motivation.
>
>     - Added Corollary 3.6: States that after successful training the student can solve all hard problems in the tree. This is, of course, the purpose of Theorem 3.5, but wasn't stated separately. (reply to "... it does not seem to be the case that the authors prove any new instances are actually tractable ...")
>
>     - Added Corollary 4.4: This formalizes the discussion after Theorem 4.2, which was in plain text, previously.
>
>     - Added Lemma 5.3: This formalizes the observation that Curriculum I is learnable.
>
>     - Added Lemmas 5.4 and 5.5. This formalizes the observation that Curriculum II and III contain global \ell_0 minimizers.
>
>     Section 4.2 is in nature a afterthought or outlook and therefore left informal. In addition a formal result would require a thorough definition of the extended tree, which would make the paper substantially more technical.
>
>     *References:* I've included a paragraph on teaching dimension and knowledge distillation.
>
> 1. *3-SAT:* I've changed the wording. Generalizations of 1-in-3-SAT are considered in the numerical experiments.
>
> 1. *Remark 3.1:* See above.
>
> 1. *(M1)-(M8):*  (M1)-(M8) are not mentioned in Section 4. Does this refer to Section 5? I've formalized the reference into a lemma and moved the reference to (M1)-(M8) with the proof to the appendix.
>
> 1. *Section 5.2:* This section motivates the choice of numerical experiments for which it is normal not to include formalized theorems and proofs. But I agree that it is helpful to  formalized several related statements later in the section (new Lemmas 5.3, 5.3 and 5.5).
>
> 1. *$\ell_0$ optimizers:*  All results are recovery results, i.e. the student learns the classes the teacher provides . In all cases $\ell_0$ optimality has to be shown separately, e.g. in the new numerical example Lemmas 5.4, 5.5. This was already discussed in multiple places, but I've made the following changes:
>
>    - Add a new Remark 2.1 that is referenced in multiple places.
>    - Add a comment/reference after each class definition, Theorem 2.5 and Corollary 3.6, where the question of $\ell_0$ optimality is relevant.
>    - Change the definition of SOLVE in Section 2. If I haven't missed anything, no theoretical statement should refer to $\ell_0$ optimality any longer, only to recovery.
>    - Added a comment after (3) in the introduction.
>
>    *Does Remark 5.2 imply that in the 1-in-3 SAT setup Xz is always ...:* No, this has to be shown separately and is not true for Curriculum I. I've added Lemmas 5.4 and 5.5 to show \ell_0 optimality for Curricula II/III.
>
> 1. *(M5):*  Rewritten.
>
> 1. *Missing definitions:* Done. I've also added a Glossary at the end of the paper.
>
>    - *Partition:* Does this refer to $K_i$ in the proof? They are defined as a partition.
>    - *preconditioner T:* The preconditioner is not required in the main body. All results in the main part use the NSP, which is invariant under preconditioning, while the proofs rely on the RIP, for which it is important. I've added a comment.
>
> 1. *(A1):*  I've included both variants into (A1) and the new (T2): The constrained solution space is the teacher's perspective, whereas the pairs (A,b) are the student's perspective.
>
> 1. *Probabilistic Quantifier on 4.2:* The former is correct, since both $X$ and the curriculum are largely random. More deterministic $X$ would be desirable but are difficult to handle with current compressed sensing theory.
>
> 1. *Dropped $A$, cross terms:* Corrected A and added some explanation for the cross terms.

---

> ### Author Response · Authors · 2023-07-31
> **Reply to Reviewer (Minor Issues)**
>
> Below are comments for the "less serious issues" in same order as in the review.
>
> 1. *Typos:* Done
> 2. *Top of page 24:* $\sqrt{2}t$ sparse, corrected.
> 3. *RIP:* (T5) assumes NSP with certainty. The probabilistic statement is in the tree construction Theorem 4.2.
> 4. *$\tau$:* Corrected
> 5. *Figure:* Done
> 6. *$C_{easy}$:* Aligned the definitions
> 7. *Related work:* Added section header
> 8. *Human learning / randomness:* Added a discussion in the introduction.

---

### Review · Reviewer_xAnM · 2023-07-17

**Summary Of Contributions:**

- This work considers the problem of learning minimally sparse solutions to underdetermined linear systems Ax=b where x is sparse. Usually, one needs certain assumptions on A like RIP property. This work considers alternative set of tractable problems which can be adapted to new situations based on prior knowledge.
- This work also looks at variations of 3-SAT with weaker assumptions.


**Audience:**

Yes

**Claims And Evidence:**

Yes

**Requested Changes:**

- The authors should clearly explain what the contributions of this work are. The idea of learning problems of increasing difficulty is already introduced in a previous work as mentioned in the paper. Do the authors extend this setting to the problem of sparse linear system solving?
- Can the authors clearly explain the setting in the beginning of the paper?
- How do the authors get around the NP hardness of the problem?
- On page1, how do the authors reach equation 2? Why does sparsity of x imply sparsity of z?
- In section 2.2, it is not clear to the me what the authors mean by easy samples and hard samples? if x is uknown, and X is unknown, how are easy x chosen to generate the easy data?

**Strengths And Weaknesses:**

- The paper looks at this interesting idea of learning a series of l0 minimization problems of increasing difficulty.
- The main weakness in my opinion is that the paper is written in a way which is very hard to understand currently. The setting and the contributions are not clear.

---

> ### Author Response · Authors · 2023-07-31
> **Reply to Reviewer**
>
> Thank you for the review. Changes in the manuscript are in blue. The replies below are in the same order as the bullet points in the review.
>
> 1.  *New Contributions:* The previous paper learns hard problems from easy problems. The new contribution is to concatenate multiple such learning episodes into a curriculum tree. I've expanded the introduction and added a section "New Contributions".
>
> 1. *Setting:* I've expanded the introduction. It provides more detail now.
>
> 1. *NP-hardness:* I've included a comment at the end of the new section "New Contributions". In short, the teacher knows the answers and the information he provides are sufficient for the student to follow.
>
> 1. *Equation (2):* Sparsity of $x$ does not imply sparsity of $z$. The second condition is only sufficient. I've expanded the explanation.
>
> 1. *Easy Samples:* I've added some explanation after the definition of SOLVE in Section 2.2. In short, in the paper easy samples are defined to be sparser than hard samples. If these are practically easier depends on the existence of SOLVE. This is delicate in Section 2 and the prior work Welper (2021). One of the main contributions of this paper is to provide a candidate for SOLVE based on the hierarchical curriculum, discussed in Section 3.

---

### Review · Reviewer_9efh · 2023-07-18

**Summary Of Contributions:**

In this work, the authors focus on addressing the problem of computing minimally sparse solutions of under-determined linear systems, which is defined as finding the solution $x\in\mathbb{R}^n$ that has the minimum number of non-zero elements, subject to the constraint that the system of equations $Ax = b$ is satisfied. Specifically, the authors consider tractable subclasses of this problem, where the sparsity of the solution is imposed through the prior knowledge that $x = Xz$, where $z$ is sparse and $X$ is known from a curriculum of easy samples and knowledge condensation at each tree node. This allows the authors to address the problem with mild assumptions on $A$, without requiring $A$ to satisfy properties like RIP or NSP.

**Audience:**

Yes

**Claims And Evidence:**

Yes

**Requested Changes:**

See the Weaknesses above.

**Strengths And Weaknesses:**

Strengths:

The idea of using the prior knowledge $x=Xz$ to find minimally sparse solutions of under-determined linear systems without requiring RIP or NSP on the matrix $A$ is of interest.

Weaknesses:

I am not familiar with some parts of this work (e.g., about SAT solving) and did not check the technical results carefully. As far as I can tell, the following are the weaknesses of this work:

1. The motivation and main contributions are not clearly presented in the current submission. The practical applications of the considered setup and the proposed algorithms are not apparent from the submission. Additionally, all the presented theorems, specifically Theorems 2.3 and 2.4, are minor modifications of Theorem 4.2 in Welper (2021). The abstract is too vague and does not clearly mention the studied problem and the main contributions of this submission.

2. The writing can be improved by providing references for methods like SOLVE, SPARSEFACTOR, instead of letting the reader to search these terms back and forth. In addition, the authors should explicitly mention that (A1) to (A4) are assumptions and briefly discuss why they are reasonable. In (A4), the Bernoulli-Subgaussian constant should be denoted with brackets for clarity.

3. Typos: p2, $N \ne NP$ should be $P \ne NP$; "form" should be "from"; p9, Remark 3.7, "The results states"; p10, Remark 4.1, "but is is"; p12, Section 4.2, "by the following Lemma" should be "by the following lemma"; Remark 4.5, "form" should be "from".

---

> ### Author Response · Authors · 2023-07-31
> **Reply to Reviewer**
>
> Thank you for the review. Changes in the manuscript are in blue. The replies below are in the same order as the bullet points in the review.
>
> 1. *All theorems are minor modifications of prior work:* Yes, in the original version all new contributions were called "Proposition", while the theorems were a summary of previous work (and cited as such). I've renamed the propositions to Theorem 3.5 and Theorem 4.2 and referenced them in the "New Contributions" section.
>
>    *Abstract/Motivation/Contributions:* I've rewritten the abstract and parts of the introduction, including a new section "New Contributions".
>
>    *Applications:* The paper is a theoretical study of how one can learn and use prior knowledge to ease computationally hard problems. While the question is practically relevant, all numerical experiments are preliminary on small problems. A more practical implementation/application is beyond the scope of the paper.
>
> 2. *References to SOLVE, SPARSEFACTOR:* I've included references when these are used in assumptions and theorems and I've added a Glossary with references to the end of the paper.
>
>     *(A1)-(A4):* I've added some more explanation, but kept them short because they are in a summary of (Welper 2021).
>
> 3. *Typos:* Done

---

### Decision · Action_Editors · 2023-08-25

**Recommendation:** Reject

**Comment:**

Broadly, it is agreed that improving L0 recovery feasibility is an interesting problem.  However, to varying degrees, the reviewers all had significant difficulty in grasping the main claims, understanding the associated assumptions, etc.  These concerns have been alleviated to some extent in the revision, but even at this stage, one reviewer finds the paper too unclear to suggest acceptance, one feels that the decision could equally go either way, and even the most positive recommendation still has very non-minor reservations.

Based on the reviewer comments, discussion, and my own reading, I will try to summarize why such concerns remain.

**Problem setup / motivation:** There remains significant difficulty in seeing where this *specific* setting would be adopted, either in other theoretical studies or in practice.  Even the very first step in Eq. (3) could be highly suboptimal (e.g., because Xz can be much less sparse than z – see the *multiplication* of two terms in Remark 2.1).  Many assumptions are introduced but some are hard to grasp or understand the implications of (e.g., the scaling laws in (T2), the relations between different sparsity levels in (T4)).

When faced with these challenges even the problem basics, it becomes even more difficult to accept certain claims that are made in passing, e.g., how learning must be done with only (A,b) (and not x) “to be plausible”, how restrictive a statement like “ideally have sparsity O(1)” is, how restrictive/reasonable the linearity relation in (T3) is, etc.

Some evidence for usefulness is given via possible utility in SAT solving.  However, SAT solving is a *very* active research area, and there seems to be a lack of evidence that the tools presented can actually benefit the wide array of existing tools available for SAT solving  (e.g., a clear corollary relating to SAT, or convincing numerical evidence).  To be clear, the authors don't explicitly claim such benefits, but the readers should still be given a better idea of what this part of the paper is providing to the general literature.

Finally, some concerns are raised about the degree of significance over the existing work of Welper (2021), though this has at least been made clearer, and I am not taking it as a reason for rejection in itself.

**Paper flow and correctness:** The paper can be challenging to navigate and verify certain claims of, e.g.:
- The discussion after (A2) refers to Remark 2.1, which in turn refers to Lemmas 5.4 and 5.5, whose discussion highlights that one of the main theorems (Theorem 4.2) doesn’t actually necessarily hold in these cases.  Several other parts then refer back to Remark 2.1 which in itself wasn’t so easy to grasp.  These sorts of things can cause major confusion.
- Eq. (21) and Appendix C refer back to Theorem 4.2, but the claims being made are not readily visible there.
- It is unclear whether the paragraph before Lemma 5.5 (including its cross-reference to Remark 5.2) is meant to establish the lemma’s correctness, as no other proof is given.
- Lemma 5.3 is a formal statement that refers to an informal description in Figure 2, with the preceding text not entirely helping (e.g., “selected rows” is vague).
- Some terminology/phrasing isn’t clear enough, e.g., what “adaptable” means, how to interpret “columns are spread into the leaf classes”, definition of “block column”, etc.
- The purpose of certain parts is less clear, e.g., ‘Human Learning’ on p3 is an abrupt change and may be too high-level/superficial; ‘Greedy Search’ on p4 is never returned to; clarity of Remark 4.1; etc.

Please note that this is far from a complete list, but rather just examples of difficulties faced.  These sorts of difficulties appear to have persisted throughout the entire paper.

**Audience:**

Some members of the machine learning community will be interested in exploiting prior knowledge in compressive sensing, and doing so in a more efficient manner than a naïve approach.  The reviewer feedback suggests that TMLR might not be the most appropriate venue, though in terms of general topic it should be a sufficient match.  However, a number of non-minor concerns remain, as outlined below.

**Claims And Evidence:**

This paper studies sparse recovery problems with prior information that the signal lies in the column-space of a known matrix.  Instead of having direct access to X, it is assume to be known to a “teacher” and taught to a “student”.  Instead of doing so in one step, this paper does so in a hierarchical manner following a tree structure.

In the initial submission, the reviewers had significant difficulties grasping what the main claims are.  This is now alleviated to some extent, but not fully.  Broadly, the claims are identifying mathematical conditions under which the teaching/learning succeeds.  The evidence is in the form of theoretical proofs (rather than experiments etc.).  The required conditions are given in (T1)-(T8) and Theorem 4.2, and some conditions are easier to grasp than others.  While the reviewers did not identify any incorrect claims, the clarity/presentation issues (discussed below) may make it difficult for readers to (i) readily grasp the main claims, and (ii) verify their correctness.